# RECTOR: Masked Region-Channel-Temporal Modeling for Affective and Cognitive Representation Learning

**Jinhan Liu** [1]  **Mahsa Shoaran** [1]

## Abstract

Affective and cognitive disorders manifest as distributed, time-varying brain network dynamics across regions, channels, and time, challenging robust representation learning from EEG/sEEG for clinical diagnosis. We propose **RECTOR** (Masked **Re**gion–**C**hannel–**T**emp**o**ral Modeling), an end-to-end self-supervised framework that unifies joint region-channel-temporal representation learning beyond fixed anatomical priors. At its core, **RECTOR-SA** is a hierarchical, block-sparse self-attention induced by Adaptive Functional Partitioning that evolves region structures from static anatomical definitions to adaptive functional regions. The self-supervision is driven by **Masked Topology and Representation Learning**, which jointly optimizes three complementary objectives: Masked Predictive Modeling, Topological Structure Modeling, and Cross-View Consistency. Across diverse benchmarks, REC-TOR sets a new state-of-the-art in EEG emotion recognition and sEEG task-engagement classification. Crucially, its strong robustness to missing channels and cross-montage generalization underscores its potential for large-scale pre-training on heterogeneous EEG/sEEG, providing interpretable insights at both region and channel levels.

## 1. Introduction

Affective disorders and cognitive impairments affect hundreds of millions worldwide, substantially diminishing quality of life and imposing a heavy socio-economic burden (World Health Organization, 2022). Despite decades of research, clinical assessment still relies on patient self-reports and behavioral observations, which are inherently

[1] Institutes of Electrical and Micro Engineering and Neuro-X, EPFL, Geneva, Switzerland. Correspondence to: Jinhan Liu <jinhan.liu@epfl.ch>.

*Proceedings of the 43rd International Conference on Machine Learning*, Seoul, South Korea. PMLR 306, 2026. Copyright 2026 by the author(s).

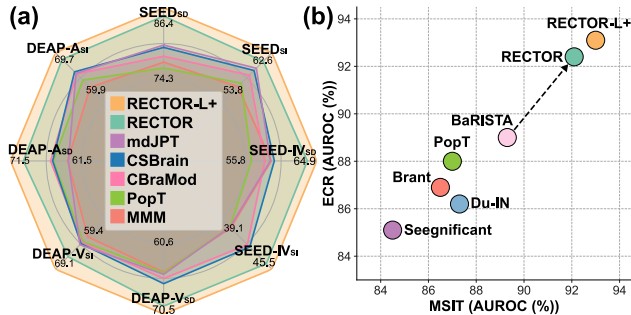

*Figure 1.* (a) RECTOR consistently outperforms leading EEG models on emotion recognition benchmarks (Weighted-F1 (%)). SD: subject-dependent; SI: subject-independent; V: valence; A: arousal. (b) RECTOR outperforms state-of-the-art models on sEEG task engagement classification.

subjective and can delay diagnosis and treatment (Kret & Ploeger, 2015; Goschke, 2014). In contrast, neurophysiological signals such as electroencephalography (EEG) and stereo-EEG (sEEG) offer direct, noninvasive/invasive windows into the brain's electrical activity. Analysis of these signals promises objective biomarkers for affective and cognitive states, enabling novel brain–computer interfaces (BCIs) and personalized interventions (Houssein et al., 2022; Drane et al., 2021; Liu et al., 2024). However, EEG/sEEG data present formidable challenges for learning affective and cognitive neural representations.

Mood and cognitive disorders arise from distributed, dynamic interactions across different brain regions, requiring models that explicitly capture region-level dynamics (Lindquist et al., 2012; Menon, 2011; Pessoa, 2017). Although recent methods have aimed to capture regional brain dynamics, they exhibit several key limitations. Many approaches using bihemispheric modeling (Li et al., 2020; Huang et al., 2021), graph neural networks (Ding et al., 2023; Ye et al., 2022; Jin et al., 2024; Qiu et al., 2023) and transformers (Zheng et al., 2024; Yi et al., 2023), attempt to learn regional embeddings from scratch but lack a dedicated region learning framework to guide this process, resulting in suboptimal representations. Crucially, these approaches rely on fixed anatomical priors, failing to capture the adaptive functional structure that evolves during cognitive processing. This rigid modeling severely limits robustness against missing channels and prevents

cross-montage generalization—two fundamental barriers to real-world clinical deployment. Furthermore, by overlooking topological structure modeling, standard methods often learn superficial shortcuts rather than structurally invariant, neurophysiologically plausible representations. Other methods oversimplify brain topology by performing population-level spatial encoding (Chau et al.; Mentzelopoulos et al., 2024), treating all channels as a single global region and thus ignoring the brain's established functional network architecture. Moreover, recent models like MMM (Yi et al., 2023) and Du-IN (Zheng et al., 2024) discard the fine-grained, channel-specific dynamics that are essential for preserving high electrophysiological fidelity.

The sparse labeling of EEG/sEEG data makes self-supervised learning (SSL) an essential paradigm (Rafiei et al., 2024; Chien et al., 2022; Li et al., 2022; Wang et al., 2023b). However, the predominant masked modeling with random masking (Zheng et al., 2024; Zhang et al., 2023; Jiang et al., 2024a;b; Wang et al., 2024) often creates a simple pretext task. This encourages models to learn superficial spatio-temporal shortcuts rather than generalizable features, limiting performance on downstream tasks (Assran et al., 2023). This risk is particularly acute for neural data, where affective and cognitive contexts induce significant spatial heterogeneity and temporal variability (Segal et al., 2023; Wei et al., 2023). Moreover, most SSL approaches (Yi et al., 2023; Jiang et al., 2024b; Wang et al., 2025; 2023a) lack explicit constraints to structure the representation space, which can limit the quality and robustness of the final embeddings.

While transformers are prevalent in SSL for EEG/sEEG, a key challenge remains in adapting self-attention (SA) to effectively capture joint spatio-temporal interactions (Rafiei et al., 2024). Current spatial encoding approaches neglect temporal dynamics (Yi et al., 2023; Jiang et al., 2024b), while sequential or criss-cross schemes learn spatial and temporal interactions in isolation (Wang et al., 2025). Although full spatio-temporal SA models dense interactions (Jiang et al., 2024a; Zhang et al., 2023), it is not only computationally inefficient due to the large number of tokens but also risks amplifying uninformative relationships while obscuring critical dependencies. Furthermore, integrating anatomical priors and brain functional dynamics into SA remains a non-trivial challenge.

To tackle these critical challenges in affective and cognitive neural representation learning, we introduce **RECTOR**: Masked **Re**gion-**C**hannel-**T**emp**or**al Modeling. RECTOR makes the following key contributions:

1. **Unified Region-Channel-Temporal Architecture.** We introduce **RECTOR-SA**, a hierarchical, block-sparse self-attention mechanism driven by **Adaptive Functional Partitioning (AFP)**. Unlike previous methods restricted to fixed anatomical priors, AFP evolves static definitions into task-specific functional regions. Combined with top-$p$ gating, it effectively prunes irrelevant connections, enabling the model to disentangle distinct region-common and channel-specific neural dynamics while serving as a learned denoiser.

2. **Masked Topology and Representation Learning (MTRL).** We propose a holistic self-supervised learning paradigm that unifies three complementary objectives within a single forward pass: (a) **Masked Predictive Modeling (MPM)** for robust local and global feature reconstruction, (b) **Topological Structure Modeling (TSM)** to explicitly regularize the learned region-channel assignments, and (c) **Cross-View Consistency (CVC)** to enforce representation invariance across diverse structural views generated by our Topology-Aware Multi-View Masking.

Across diverse benchmarks, RECTOR sets a new state-of-the-art in EEG emotion recognition and sEEG task engagement classification by consistently outperforming leading supervised and self-supervised methods. This superior performance is coupled with a remarkable computational efficiency in spatio-temporal SA. Crucially, it demonstrates superior zero-shot robustness to missing channels and cross-montage generalization, underscoring its potential for large-scale pre-training on heterogeneous neurophysiological data. Moreover, RECTOR offers deep interpretability through multi-level visualizations of neural dynamics, which align with established neurophysiology underlying cognitive conditions. We further show that AFP learns stable and spatially coherent refinements beyond the anatomical prior, rather than simply preserving the initialization. This trifecta of accuracy, efficiency, and interpretability positions RECTOR as a powerful framework for advancing neurocognitive diagnostics and enabling adaptive interventions.

## 2. Methodology

### 2.1. Problem Setting and Formulation

We model an EEG/sEEG trial as $\mathbf{X}_{\text{in}} \in \mathbb{R}^{C \times S}$, where $C$ denotes the number of recording channels and $S$ the number of time samples. The problem is formulated as a classification task with a label $y$ inferred from $\mathbf{X}_{\text{in}}$. Let $L$ be the patch length and $T = S/L$, and we define $\mathcal{U}_L : \mathbb{R}^{C \times (TL)} \to \mathbb{R}^{(CT) \times L}$, then $\mathbf{X}_{\text{patch}} = \mathcal{U}_L(\mathbf{X}_{\text{in}})$. The patches are projected to $d$-dimensional token space by $\mathbf{X}^{\text{C}} = \mathbf{X}_{\text{patch}} \mathbf{W}_e \in \mathbb{R}^{CT \times d}$, where $\mathbf{W}_e \in \mathbb{R}^{L \times d}$. This $\mathbf{X}^{\text{C}}$ is the input channel-time tokens processed by RECTOR.

### 2.2. RECTOR

**Overview** Masked **Re**gion-**C**hannel-**T**emp**or**al Modeling (**RECTOR**) introduces a holistic self-supervised learning (SSL) paradigm for modeling region-channel-

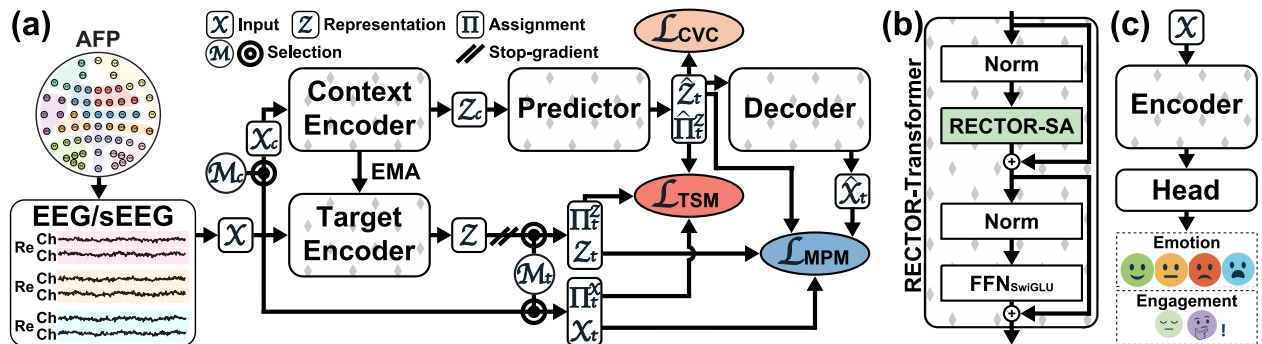

*Figure 2.* **Overview of RECTOR.** (a) The SSL pipeline utilizes Adaptive Functional Partitioning (AFP) and three objectives: **Masked Predictive Modeling (MPM)**, **Topological Structure Modeling (TSM)**, and **Cross-View Consistency (CVC)**. (b) The Transformer backbone features **RECTOR-SA** in the context/target encoders, predictor, and decoder. (c) The fine-tuning pipeline with the pre-trained encoder and a decoding head for downstream emotion recognition and task engagement classification.

temporal representations. This requires a hierarchical transformer architecture and a unified **Masked Topology and Representation Learning** framework, comprising four co-designed components: (1) **RECTOR-SA** (Fig. 2(b)), a hierarchical, block-sparse self-attention building block with adaptive region–channel topology; (2) **Masked Predictive Modeling** (Fig. 2(a)), a predictive objective in input and latent spaces to learn rich representations; (3) **Topological Structure Modeling** (Fig. 2(a)), a structural objective that captures stable functional topology from partial contexts; (4) **Cross-View Consistency** (Fig. 2(a)), an invariance objective to learn robust semantic representations across diverse views. After SSL pretraining for learning generalized region-channel-temporal representations, the context encoder is fine-tuned for downstream tasks (Fig. 2(c)), including emotion recognition and task-engagement classification.

**Hierarchical Tokenization** Given the input channel-time tokens $\mathbf{X}^{\mathrm{C}} \in \mathbb{R}^{CT \times d}$, we explicitly model the brain's hierarchical structure by introducing region tokens $\mathbf{X}^{\mathrm{R}}$. Let $\mathbf{\Pi}^{\mathbf{X}} \in [0,1]^{C \times R}$ be the soft channel-to-region assignment in the input-space (detailed in Section 2.3) and $\mathbf{E}^{\mathrm{R}} \in \mathbb{R}^{R \times d}$ be the region identity embeddings. $\mathbf{X}^{\mathrm{R}}$ are initialized by aggregating channel information: $\forall t, \ [\mathbf{X}^{\mathrm{R}}]_{t,:} = \mathbf{\Pi}^{\mathbf{X}\top}[\mathbf{X}^{\mathrm{C}}]_{t,:} + \mathbf{E}^{\mathrm{R}}$. These are concatenated with $\mathbf{X}^{\mathrm{C}}$ to form the hierarchical sequence $\mathbf{X} = [\mathbf{X}^{\mathrm{C}}; \mathbf{X}^{\mathrm{R}}] \in \mathbb{R}^{((C+R)T) \times d}$ fed to RECTOR.

**Architecture & Pipeline** RECTOR comprises four RECTOR-Transformer modules (Appendix B.5) using **RECTOR-SA** (Fig. 2(a) and 2(b)): a context encoder $f(\,\cdot\,;\theta)$, a target encoder $\tilde{f}(\,\cdot\,;\tilde{\theta})$ (EMA replica of $f(\,\cdot\,;\theta)$), predictor $g(\,\cdot\,;\psi)$, and decoder $h(\,\cdot\,;\phi)$. The SSL pipeline in Fig. 2(a) illustrates that $\mathbf{X}$ is first partitioned into a visible context view $\mathbf{X}_c$ and masked target views $\mathbf{X}_t$ with a selection matrix for target tokens $\mathbf{M}_t$. The context encoder maps $\mathbf{X}_c$ to latent representations $\mathbf{Z}_c$, while the target encoder

processes $\mathbf{X}$ to generate stable target representations $\mathbf{Z}_t$:

$$\mathbf{Z}_c = f(\mathbf{X}_c, \mathbf{E}_c; \theta), \quad \mathbf{Z}_t = \mathbf{M}_t \, \tilde{f}\left(\mathbf{X}, \mathbf{E}; \tilde{\theta}\right) \tag{1}$$

where $\mathbf{E}_c$ and $\mathbf{E}$ are the positional embeddings of the context and global view. Conditioned on target positional embeddings $\mathbf{E}_t$, the predictor infers target representations $\hat{\mathbf{Z}}_t$ from $\mathbf{Z}_c$, and the decoder reconstructs target input signal $\hat{\mathbf{X}}_t$:

$$\hat{\mathbf{Z}}_t = g(\mathbf{Z}_c, \mathbf{E}_t; \psi), \quad \hat{\mathbf{X}}_t = h(\hat{\mathbf{Z}}_t, \mathbf{E}_t; \phi) \tag{2}$$

The masked assignment matrices in the input space ($\mathbf{\Pi}_t^{\mathbf{X}}$), predicted in the latent space ($\hat{\mathbf{\Pi}}_t^{\mathbf{Z}}$), and from the target encoder's latent space ($\tilde{\mathbf{\Pi}}_t^{\mathbf{Z}}$) can be computed from $\mathbf{X}_t, \hat{\mathbf{Z}}_t, \mathbf{Z}, \mathbf{M}_t$ via Eq. 4 (detailed in Section 2.4). Together $\mathbf{X_t}, \hat{\mathbf{X}}_t, \mathbf{Z_t}, \hat{\mathbf{Z}_t}$ are used for SSL via **Masked Predictive Modeling** and **Cross-View Consistency**, while $\mathbf{\Pi}_t^{\mathbf{X}}, \hat{\mathbf{\Pi}}_t^{\mathbf{Z}}, \tilde{\mathbf{\Pi}}_t^{\mathbf{Z}}$ are used for **Topological Structure Modeling**. In the fine-tuning pipeline (Fig. 2(c)), $\mathbf{X}$ is forwarded through the pre-trained encoder and a decoding head for downstream classification optimized with cross-entropy.

### 2.3. Region-Channel-Temporal Self-Attention

**Motivation** Standard self-attention typically treats EEG/sEEG channels as unstructured sequences, obscuring the brain's latent regional organization and failing to model explicit region-level interactions. This is exacerbated by inter-subject variability, where a fixed atlas-defined partition may misalign with individual functional coupling. To address this, we propose Region-Channel-Temporal Self-Attention (**RECTOR-SA**) (Fig. 3), a hierarchical attention mechanism that learns dynamic regional structures to induce block-sparse region–channel–temporal attention. RECTOR-SA disentangles neural representations into parallel streams of region tokens (region-common, high-level dynamics) and channel tokens (channel-specific, fine-grained dynamics). We initialize region-channel partitions with anatomical priors but adapt them via learned assignments to capture functional evidence. A data-driven

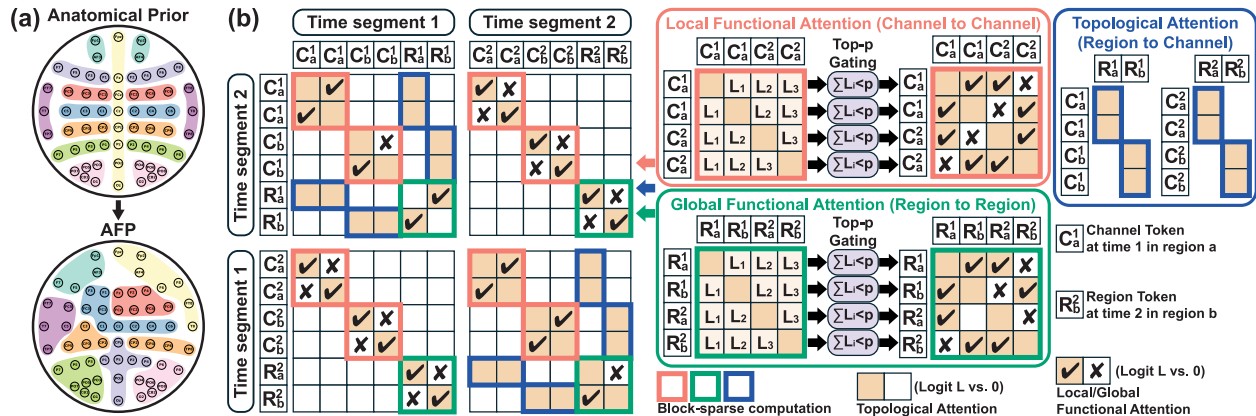

Figure 3. **RECTOR-SA.** (a) **Adaptive Functional Partitioning (AFP).** (Top) Anatomical prior based on standard electrode layouts. (Bottom) Learned region-channel assignment, where channels are dynamically grouped into functional regions rather than fixed anatomical lobes. (b) **Anatomical Attention**, **Local Functional Attention**, and **Global Functional Attention** are used for modeling region-channel-temporal interactions with block-sparse computation. A top-$p$ gating operator dynamically prunes weak connections (marked with ×).

top-$p$ gating further acts as a denoiser, pruning spurious connections to focus on salient topological interactions.

**Masked Attention** Let $\mathbf{X_Q}$, $\mathbf{X_K}$, $\mathbf{X_V}$ denote the input token sequences, and we compute $\mathbf{Q} = \mathbf{X_Q W_Q}$, $\mathbf{K} = \mathbf{X_K W_K}$, $\mathbf{V} = \mathbf{X_V W_V}$, $\mathbf{L} = \mathbf{QK}^\top/\sqrt{d}$. Given a binary mask $\mathbf{S} = \{0,1\}^{N \times N}$, we convert it to an additive mask $\mathcal{B}(\mathbf{S})$ by setting $\mathcal{B}(\mathbf{S})_{ij} = 0$ if $\mathbf{S}_{ij} = 1$ and $\mathcal{B}(\mathbf{S})_{ij} = -\infty$ otherwise. The masked attention can be formulated as:

$$\mathbf{L}_{\text{mask}} = \text{AttnLogit}(\mathbf{X_Q}, \mathbf{X_K}, \mathbf{S}) = \mathbf{L} + \mathcal{B}(\mathbf{S})$$
$$\mathbf{Z} = \text{Attn}(\mathbf{L}_{\text{mask}}, \mathbf{X_V}) = \text{softmax}(\mathbf{L}_{\text{mask}})\mathbf{V} \quad (3)$$

**Adaptive Functional Partitioning (AFP)** To capture dynamic functional couplings beyond anatomical boundaries, AFP partitions the sensor array into latent functional regions. Given the channel tokens $\mathbf{U}$, $R$ learnable region prototypes $\mathbf{P} \in \mathbb{R}^{R \times d}$, an anatomical prior assignment matrix $\mathbf{S}^{\text{anat}} \in \mathbb{R}^{C \times R}$, and a projection matrix $\mathbf{W} \in \mathbb{R}^{d \times d}$, we define a channel-to-region adaptive assignment head:

$$\mathbf{\Lambda}(\mathbf{U}, \mathbf{W}) = \text{softmax}((\frac{\bar{\mathbf{U}}\mathbf{W}\mathbf{P}^\top}{\sqrt{d}} + \alpha(t)\mathbf{S}^{\text{anat}} + \mathbf{G})/\tau) \quad (4)$$

where $\bar{\mathbf{U}} = \frac{1}{T}\sum_{t=1}^{T}\mathbf{U}_t$ is the temporal token average, $\alpha(t)$ is an annealing factor over training, $\tau$ is a temperature parameter, and $\mathbf{G}_{i,j} \sim \text{Gumbel}(0,1)$. Thus, the soft ($\mathbf{\Pi^X}$) and hard ($\mathbf{S^X}$) assignment matrices for input channel tokens $\mathbf{X^C}$ are formulated as $\mathbf{\Pi^X} = \mathbf{\Lambda}(\mathbf{X^C}, \mathbf{W_\Pi^X})$ and $\mathbf{S^X} = \text{OneHot}(\text{argmax}_r(\mathbf{\Pi^X})_{:,r}$, where $\mathbf{W_\Pi^X}$ is the input-space assignment projection. For end-to-end training with discrete topology, we employ the *Straight-Through Gumbel-Softmax Estimator*: $\mathbf{S}_{\text{STE}}^{\mathbf{X}} = \mathbf{S^X} + \text{sg}(\mathbf{\Pi^X} - \mathbf{S^X})$, where $\text{sg}(\cdot)$ is the stop-gradient operator.

For EEG, we construct $\mathbf{S}^{\text{anat}}$ by grouping International 10-20 sensors into 11 regions as in Fig. 3(b) (Alarcao & Fonseca, 2017). For sEEG, we assign channels to 30 regions using the Electrode Labeling Algorithm (Peled et al., 2017)

(Appendix E for detailed maps). As $\alpha(t)$ decays, the partition evolves from these static anatomical definitions to data-driven functional regions.

**RECTOR-SA** Instead of computing the dense attention on the full token sequence $[\mathbf{X^C}; \mathbf{X^R}] \in \mathbb{R}^{(C+R)T \times d}$, RECTOR-SA factorizes it into three sparse, topology-aware mechanisms and executes them in parallel blocks: **Topological Attention**, **Local Functional Attention**, and **Global Functional Attention**, each instantiated by selecting subsequences $\mathbf{X_Q}$, $\mathbf{X_K}$, $\mathbf{X_V}$ and applying a binary mask $\mathbf{S}$.

**Topological Attention** (Fig 3(a)) We model region–channel interactions via a sparse topology induced by AFP. We compute two cross-attention directions:

$$\mathbf{L}_{\text{topo}}^{R \to C} = \text{AttnLogit}(\mathbf{X^C}, \mathbf{X^R}, \mathbf{S}_{\text{topo}})$$
$$\mathbf{L}_{\text{topo}}^{C \to R} = \text{AttnLogit}(\mathbf{X^R}, \mathbf{X^C}, \mathbf{S}_{\text{topo}}^\top) \quad (5)$$

where $\mathbf{S}_{\text{topo}} = \mathbf{S^X} \otimes \mathbf{I}_T \in \{0,1\}^{(CT) \times (RT)}$, and $\otimes$ denotes Kronecker product. Thus, each channel attends only to its assigned region at the same time slice, and vice versa.

**Local Functional Attention** (Fig 3(a)) Local functional attention models fine-grained channel interactions *within* each assigned region over time. Masked by channel co-membership $\mathbf{S}_{\text{loc}} = \mathbf{S^X}\mathbf{S^X}^\top \otimes \mathbf{1}_{T \times T} \in \{0,1\}^{(CT) \times (CT)}$ and dynamic top-$p$ gating:

$$\mathbf{L}_{\text{loc}} = \text{AttnLogit}(\mathbf{X^C}, \mathbf{X^C}, \mathbf{S}_{\text{loc}} \odot \mathcal{G}_p(\mathbf{L^C})) \quad (6)$$

$\mathcal{G}_p(\cdot)$ applies row-wise top-$p$ gating (Appendix B.4) to the logits $\mathbf{L^C}$ computed from $\mathbf{X^C}$, pruning spurious interactions. In practice, $\mathbf{S}_{\text{loc}}$ is block-diagonal by region, thus $\mathbf{L}_{\text{loc}}$ is computed independently within each region block.

**Global Functional Attention** (Fig 3(a)) Global functional attention captures high-level coordination among regions:

$$\mathbf{L}_{\text{glob}} = \text{AttnLogit}(\mathbf{X^R}, \mathbf{X^R}, \mathcal{G}_p(\mathbf{L^R})) \quad (7)$$

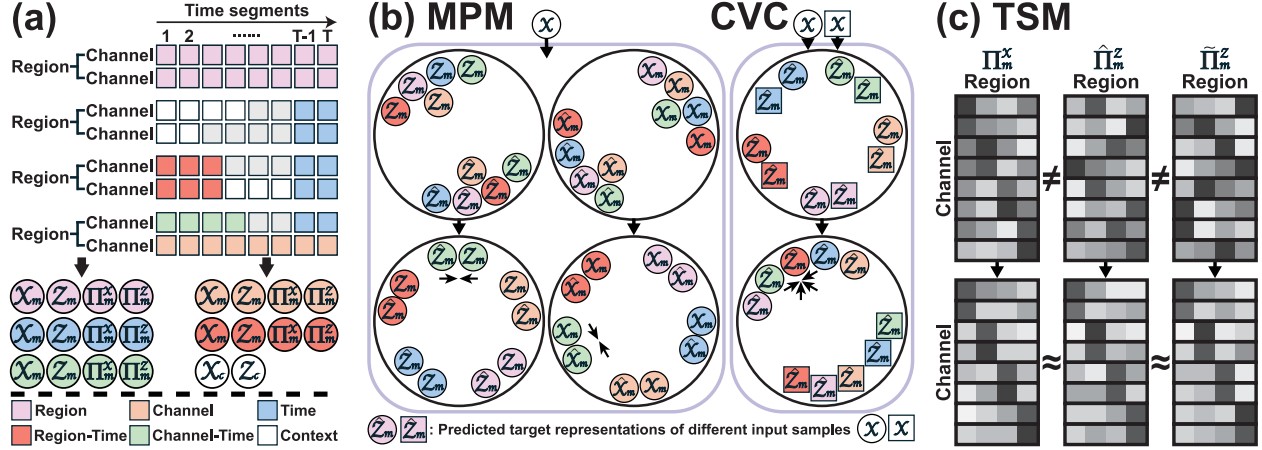

*Figure 4.* **Overview of Masked Topology and Representation Learning.** (a) **Topology-Aware Multi-View Masking** generates one context view and five target views from distinct structural domains. (b) RECTOR optimizes **Masked Predictive Modeling (MPM)** for predictive representation learning in the input and latent spaces, and **Cross-View Consistency (CVC)** for invariant representation learning across diverse intra-sample views. (c) **Topological Structure Modeling (TSM)** explicitly predicts and aligns the learned region-channel assignments to preserve topological coherence.

where $\mathbf{L}^{\mathrm{R}}$ denotes the attention logits computed from $\mathbf{X}^{\mathrm{R}}$.

**Fusion** Let $\mathbf{X_V} = [\mathbf{X}^{\mathrm{C}}; \mathbf{X}^{\mathrm{R}}]$, $\mathbf{L}^{\mathrm{C}}_{\mathrm{mask}} = [\mathbf{L}_{\mathrm{loc}} \ \mathbf{L}^{R\to C}_{\mathrm{topo}}] \in \mathbb{R}^{(CT\times(C+R)T)}$ and $\mathbf{L}^{\mathrm{R}}_{\mathrm{mask}} = [\mathbf{L}^{R\to C}_{\mathrm{topo}} \ \mathbf{L}_{\mathrm{glob}}] \in \mathbb{R}^{(RT\times(C+R)T)}$, these blocks can be fused as:

$$\mathbf{Z}^{\mathrm{C}} = \mathrm{Attn}(\mathbf{L}^{\mathrm{C}}_{\mathrm{mask}}, \mathbf{X_V}), \quad \mathbf{Z}^{\mathrm{R}} = \mathrm{Attn}(\mathbf{L}^{\mathrm{R}}_{\mathrm{mask}}, \mathbf{X_V}) \quad (8)$$

In practice, $\mathbf{L}^{\mathrm{C}}_{\mathrm{mask}}$ is block-sparse so $\mathbf{Z}^{\mathrm{C}}$ can be computed independently within each region block. The final output of RECTOR-SA is $\mathbf{Z} = [\mathbf{Z}^{\mathrm{C}}; \mathbf{Z}^{\mathrm{R}}]$. This design yields block-sparse computation and avoids quadratic attention over all region-channel–time tokens while preserving multi-scale structure. See Appendix B.5 for the RECTOR-Transformer block using multi-head RECTOR-SA.

### 2.4. Masked Topology and Representation Learning

Standard SSL methods typically focus solely on local signal reconstruction while neglecting the global network dynamics. To bridge this gap, we propose **Masked Topology and Representation Learning**, a unified framework that jointly learns three complementary objectives: predicting missing signals and representations, inferring stable functional networks, and ensuring semantic consistency across views.

**Topology-Aware Multi-View Masking** We construct a structured pretext task on the hierarchical neurophysiological space by splitting $\mathbf{X}^{\mathrm{C}}$ into a visible context view and multiple disjoint masked target views. Let $\mathbf{M}_c, \{\mathbf{M}_m\}_{m\in\mathcal{M}}$ be row-one-hot selection matrices for selecting context and target tokens for domain $m \in \mathcal{M}$, and selections are disjoint across views. The context and target inputs are $\mathbf{X}_c = [\mathbf{M}_c\mathbf{X}^{\mathrm{C}}; \mathbf{X}^{\mathrm{R}}]$ and $\mathbf{X}_m = [\mathbf{M}_m\mathbf{X}^{\mathrm{C}}; \mathbf{X}^{\mathrm{R}}]$. Crucially, the targets are sampled by drawing $\mathbf{M}_m$ from five topological domains $\mathcal{M} = \{\mathrm{r,c,t,rt,ct}\}$: region (r), chan-

nel (c), time (t), region-time (rt), and channel-time (ct); $\mathbf{M}_c$ is then defined for selecting the residual tokens. This masking scheme provides the required views for Masked Predictive Modeling, Topological Structure Modeling, and Cross-View Consistency.

**Masked Predictive Modeling (MPM)** Our goal is to learn representations that are both *semantically rich* for downstream tasks and *grounded in signal-level detail*. To this end, we optimize a masked predictive objective that couples an input reconstruction term with a representation alignment term. Concretely, for each target view $m \in \mathcal{M}$, the model predicts masked representations $\hat{\mathbf{Z}}_m$ and decodes masked inputs $\hat{\mathbf{X}}_m$, which are supervised by original target representation $\mathbf{Z}_m$ and input $\mathbf{X}_m$, respectively (Fig. 4(b)). The representation alignment encourages semantically predictive features but is prone to degenerate solutions, while the input reconstruction anchors the encoder to remain informative and mitigates representational collapse (Appendix B.3). With the number of target views $|\mathcal{M}|$, our MPM loss can be formulated as:

$$\mathcal{L}_{\mathrm{MPM}} = \frac{1}{|\mathcal{M}|} \sum_{m\in\mathcal{M}} \mathcal{L}^{(m)}_{\mathrm{MPM}}$$
$$\mathcal{L}^{(m)}_{\mathrm{MPM}} = \|\hat{\mathbf{X}}_m - \mathbf{X}_m\|_2^2 + \|\hat{\mathbf{Z}}_m - \mathrm{sg}\,(\mathbf{Z}_m)\|_2^2 \quad (9)$$

**Topological Structure Modeling (TSM)** Beyond predicting masked signals, we require the representations to be predictive for a stable functional topology. Therefore, we optimize TSM that unifies a masked topology prediction term with an assignment matching term. In the first term, the model predicts the masked assignment in the latent space $\hat{\mathbf{\Pi}}^{\mathbf{Z}}_m = \Lambda(\hat{\mathbf{Z}}_m, \mathbf{W}^{\mathbf{Z}}_{\mathbf{\Pi}})$, supervised by the stable assignment $\tilde{\mathbf{\Pi}}^{\mathbf{Z}}_m = \mathbf{M}_m\Lambda(\mathbf{Z}, \mathbf{W}^{\mathbf{Z}}_{\mathbf{\Pi}})$ from the target encoder, where $\mathbf{W}^{\mathbf{Z}}_{\mathbf{\Pi}}$ is the latent-space assignment projection (Eq. 4). This compels the model to infer global topology from partial context. Simultaneously, the assignment match-

*Table 1.* Classification performance on SEED, SEED-IV, and DEAP (w-F1: Weighted-F1 (%); BAcc: Balanced Accuracy (%)). SD: Subject Dependent; SI: Subject Independent. **BOLD**/UNDERLINE indicate the best/second-best results. * denotes it significantly outperforms the second-best model.

| Model | SEED | | | | SEED-IV | | | | DEAP-Valence | | | | DEAP-Arousal | | | |
|---|---|---|---|---|---|---|---|---|---|---|---|---|---|---|---|---|
| | SD | | SI | | SD | | SI | | SD | | SI | | SD | | SI | |
| | w-F1 | BAcc | w-F1 | BAcc | w-F1 | BAcc | w-F1 | BAcc | w-F1 | BAcc | w-F1 | BAcc | w-F1 | BAcc | w-F1 | BAcc |
| TSception | $74.5_{\pm11.8}$ | $74.2_{\pm12.0}$ | $49.2_{\pm12.1}$ | $48.9_{\pm12.2}$ | $54.3_{\pm14.0}$ | $52.7_{\pm14.6}$ | $35.5_{\pm08.1}$ | $33.6_{\pm08.2}$ | $62.0_{\pm14.5}$ | $59.8_{\pm14.9}$ | $61.5_{\pm15.3}$ | $59.9_{\pm15.3}$ | $62.9_{\pm16.2}$ | $60.6_{\pm15.6}$ | $61.3_{\pm16.0}$ | $59.7_{\pm16.3}$ |
| MMM | $77.2_{\pm10.5}$ | $77.0_{\pm10.8}$ | $56.7_{\pm10.8}$ | $56.3_{\pm10.9}$ | $59.1_{\pm12.6}$ | $58.0_{\pm12.2}$ | $40.3_{\pm07.5}$ | $39.5_{\pm07.6}$ | $64.7_{\pm12.4}$ | $62.8_{\pm12.3}$ | $62.9_{\pm13.7}$ | $61.2_{\pm13.4}$ | $63.5_{\pm15.5}$ | $60.7_{\pm15.2}$ | $63.0_{\pm16.3}$ | $61.4_{\pm15.3}$ |
| PGCN | $76.8_{\pm12.1}$ | $76.7_{\pm12.2}$ | $56.6_{\pm10.6}$ | $56.1_{\pm10.3}$ | $58.4_{\pm13.3}$ | $56.9_{\pm13.4}$ | $40.2_{\pm07.7}$ | $39.0_{\pm07.4}$ | $62.8_{\pm12.9}$ | $62.5_{\pm14.0}$ | $61.6_{\pm14.0}$ | $64.5_{\pm15.5}$ | $62.8_{\pm14.5}$ | $64.3_{\pm15.1}$ | $64.2_{\pm15.1}$ | $64.1_{\pm15.5}$ |
| EmT | $76.2_{\pm10.6}$ | $76.1_{\pm10.2}$ | $57.2_{\pm10.8}$ | $57.3_{\pm10.4}$ | $56.5_{\pm12.9}$ | $55.1_{\pm12.1}$ | $39.9_{\pm08.4}$ | $38.4_{\pm08.2}$ | $64.7_{\pm13.2}$ | $62.4_{\pm13.6}$ | $63.0_{\pm14.3}$ | $61.6_{\pm14.4}$ | $64.0_{\pm16.4}$ | $62.4_{\pm16.5}$ | $64.2_{\pm16.5}$ | $63.3_{\pm16.3}$ |
| mdJPT | $80.1_{\pm09.7}$ | $80.1_{\pm10.1}$ | $59.9_{\pm09.7}$ | $59.4_{\pm09.4}$ | $59.0_{\pm12.2}$ | $58.3_{\pm12.3}$ | $40.2_{\pm07.9}$ | $39.6_{\pm08.2}$ | $65.0_{\pm12.4}$ | $63.3_{\pm13.8}$ | $64.4_{\pm12.8}$ | $63.5_{\pm12.6}$ | $63.4_{\pm15.2}$ | $61.9_{\pm15.8}$ | $65.7_{\pm14.8}$ | $63.8_{\pm14.9}$ |
| Conformer | $66.6_{\pm12.6}$ | $66.5_{\pm13.0}$ | $51.0_{\pm11.9}$ | $51.1_{\pm11.3}$ | $55.6_{\pm14.3}$ | $54.1_{\pm14.2}$ | $37.5_{\pm08.4}$ | $36.5_{\pm07.9}$ | $64.3_{\pm14.3}$ | $62.5_{\pm12.6}$ | $62.8_{\pm15.4}$ | $61.5_{\pm13.6}$ | $64.5_{\pm16.1}$ | $63.3_{\pm15.1}$ | $62.0_{\pm16.5}$ | $61.0_{\pm15.5}$ |
| LGGNet | $68.6_{\pm11.4}$ | $68.6_{\pm11.4}$ | $50.7_{\pm11.0}$ | $50.5_{\pm10.8}$ | $50.7_{\pm13.6}$ | $49.9_{\pm13.3}$ | $36.8_{\pm08.2}$ | $35.3_{\pm09.4}$ | $64.5_{\pm13.5}$ | $63.2_{\pm13.5}$ | $61.1_{\pm15.7}$ | $59.8_{\pm13.8}$ | $62.5_{\pm15.6}$ | $62.1_{\pm15.9}$ | $62.2_{\pm15.8}$ | $61.5_{\pm15.2}$ |
| CBraMod | $78.2_{\pm10.2}$ | $78.0_{\pm10.4}$ | $58.7_{\pm09.6}$ | $58.9_{\pm09.4}$ | $58.3_{\pm12.7}$ | $57.0_{\pm12.3}$ | $42.4_{\pm06.5}$ | $42.3_{\pm07.0}$ | $65.6_{\pm11.4}$ | $64.2_{\pm12.5}$ | $64.0_{\pm12.2}$ | $63.1_{\pm12.8}$ | $65.9_{\pm14.1}$ | $63.9_{\pm13.1}$ | $65.6_{\pm14.5}$ | $64.2_{\pm14.5}$ |
| PopT | $76.2_{\pm11.4}$ | $75.7_{\pm11.2}$ | $57.3_{\pm10.1}$ | $57.0_{\pm10.3}$ | $56.6_{\pm13.0}$ | $55.2_{\pm13.3}$ | $40.2_{\pm07.5}$ | $40.6_{\pm07.7}$ | $64.8_{\pm12.3}$ | $63.5_{\pm12.3}$ | $63.8_{\pm12.6}$ | $62.8_{\pm12.9}$ | $65.5_{\pm15.4}$ | $64.1_{\pm15.4}$ | $64.4_{\pm15.8}$ | $63.2_{\pm15.1}$ |
| CSBrain | $79.7_{\pm10.1}$ | $79.4_{\pm10.4}$ | $59.5_{\pm09.8}$ | $59.1_{\pm10.1}$ | $59.5_{\pm13.0}$ | $59.2_{\pm13.1}$ | $42.7_{\pm06.9}$ | $41.1_{\pm07.3}$ | $66.3_{\pm11.2}$ | $64.8_{\pm12.6}$ | $64.2_{\pm12.3}$ | $62.9_{\pm12.3}$ | $65.6_{\pm14.8}$ | $63.6_{\pm14.4}$ | $66.0_{\pm15.0}$ | $64.9_{\pm14.7}$ |
| **RECTOR** | $\mathbf{85.0^*_{\pm08.5}}$ | $\mathbf{84.7^*_{\pm08.7}}$ | $\mathbf{61.1^*_{\pm08.5}}$ | $\mathbf{61.2^*_{\pm08.8}}$ | $\mathbf{63.7^*_{\pm11.0}}$ | $\mathbf{62.4^*_{\pm11.4}}$ | $\mathbf{44.8^*_{\pm06.7}}$ | $\mathbf{43.7^*_{\pm06.8}}$ | $\mathbf{69.4^*_{\pm09.5}}$ | $\mathbf{67.9^*_{\pm10.1}}$ | $\mathbf{66.7^*_{\pm10.9}}$ | $\mathbf{65.8^*_{\pm11.2}}$ | $\mathbf{69.7^*_{\pm12.2}}$ | $\mathbf{67.8^*_{\pm12.4}}$ | $\mathbf{68.4^*_{\pm12.1}}$ | $\mathbf{67.5^*_{\pm13.3}}$ |

ing enforces consistency between $\hat{\mathbf{\Pi}}_m^{\mathbf{Z}}$ and the input-space $\mathbf{\Pi}_m^{\mathbf{X}} = \Lambda(\mathbf{X}_m, \mathbf{W}_{\Pi}^{\mathbf{X}})$ used by AFP, which ensures the latent topology remains faithful to the input partitioning. By using row-wise cross-entropy (CE), the TSM loss is:

$$\mathcal{L}_{\text{TSM}} = \frac{1}{|\mathcal{M}|} \sum_{m \in \mathcal{M}} \mathcal{L}_{\text{TSM}}^{(m)} \tag{10}$$
$$\mathcal{L}_{\text{TSM}}^{(m)} = \text{CE}(\text{sg}(\tilde{\mathbf{\Pi}}_m^{\mathbf{Z}}), \hat{\mathbf{\Pi}}_m^{\mathbf{Z}}) + \text{CE}(\text{sg}(\hat{\mathbf{\Pi}}_m^{\mathbf{Z}}), \mathbf{\Pi}_m^{\mathbf{X}})$$

**Cross-View Consistency (CVC)** Each masked target constitutes an intra-sample view of the EEG/sEEG trial, sharing the same global context. To encourage the learning of robust semantic representations that are consistent across multiple views, we introduce CVC that attracts representations from diverse views of the same input (Fig. 4(b)). Concretely, for a set of predicted targets $\{\hat{\mathbf{Z}}_m\}_{m \in \mathcal{M}}$ from a single sample, we apply mean pooling by $\check{\mathbf{z}}_m = \sum_{n=1}^{N_m} [\hat{\mathbf{Z}}_m]_{n,:}/N_m$ ($N_m$ is the number of masked tokens) and project them with $\mathbf{W}_p$. The CVC loss is formulated as:

$$\mathcal{L}_{\text{CVC}} = \frac{2}{|\mathcal{M}|(|\mathcal{M}| - 1)} \sum_{m_1 < m_2} \|\check{\mathbf{z}}_{m_1} \mathbf{W}_p - \check{\mathbf{z}}_{m_2} \mathbf{W}_p\|_2^2 \tag{11}$$

**Unified Objective** We jointly optimize the above components to learn representations that are (1) predictive of masked neural signals and representations ($\mathcal{L}_{\text{MPM}}$), (2) topologically predictive and consistent with the learned functional structure ($\mathcal{L}_{\text{TSM}}$), and (3) invariant across intra-sample masked views ($\mathcal{L}_{\text{CVC}}$). We additionally incorporate Region-Channel Regularization ($\mathcal{L}_{\text{RCReg}}$), which explicitly disentangles region and channel tokens to encode region-common and channel-specific information by enforcing variance and covariance constraints. This avoids trivial region representations and prevents topological collapse (Appendix B.2). The total pretraining objective is:

$$\mathcal{L} = \mathcal{L}_{\text{MPM}} + \lambda_1 \mathcal{L}_{\text{TSM}} + \lambda_2 \mathcal{L}_{\text{CVC}} + \lambda_3 \mathcal{L}_{\text{RCReg}} \tag{12}$$

## 3. Experiments

### 3.1. Experimental Setup

**Datasets** We evaluate RECTOR on three publicly available EEG emotion recognition benchmarks: **SEED** (Zheng & Lu, 2015) (three-valence classification), **SEED-IV** (Zheng et al., 2018) (four-valence classification), and **DEAP** (Koelstra et al., 2011) (binary high/low valence & arousal classifications), and on two sEEG task engagement datasets, **MSIT** and **ECR** (Provenza et al., 2019) (binary rest/task classifications). For SEED, SEED-IV, and DEAP, we follow both subject-dependent (SD) and subject-independent (SI) evaluation protocols; for MSIT and ECR, we adhere to the SD paradigm (see Appendix D for details on the datasets).

**Baselines** In EEG emotion recognition, we benchmark against leading task-specific models: **TSception** (Ding et al., 2022), **MMM** (Yi et al., 2023), **PGCN** (Jin et al., 2024), **EmT** (Ding et al., 2025), **mdJPT**(Zhang et al., 2025), as well as general-purpose approaches: **Conformer** (Song et al., 2022), **LGGNet** (Ding et al., 2023), **CBraMod** (Wang et al., 2025), and **CSBrain**(Zhou et al., 2025). In sEEG cognitive-state decoding, we compare against the supervised baseline **Seegnificant** (Mentzelopoulos et al., 2024), and leading self-supervised methods: **Brant** (Zhang et al., 2023), **Du-IN** (Zheng et al., 2024), **BaRISTA**(Oganesian et al., 2025). The self-supervised **PopT** (Chau et al.) is included in both EEG and sEEG experiments. **LUNA** (Döner et al., 2025) is chosen as a baseline for EEG cross-montage analysis. (Appendix A for details on baseline comparisons).

*Table 2.* Classification performance on MSIT and ECR (AUROC (%); BAcc: Balanced Accuracy (%)). **BOLD**/UNDERLINE indicates the best/second-best results. * denotes it significantly outperforms the second-best model.

| Model | MSIT | | ECR | |
|---|---|---|---|---|
| | AUROC | BAcc | AUROC | BAcc |
| Seegnificant | $84.5_{\pm06.9}$ | $75.0_{\pm09.6}$ | $85.1_{\pm06.9}$ | $76.1_{\pm09.3}$ |
| Brant | $86.5_{\pm06.3}$ | $78.4_{\pm08.5}$ | $86.9_{\pm05.9}$ | $77.9_{\pm08.8}$ |
| Du-IN | $87.3_{\pm05.4}$ | $78.5_{\pm07.9}$ | $86.2_{\pm05.9}$ | $77.9_{\pm08.3}$ |
| PopT | $87.0_{\pm05.8}$ | $77.5_{\pm07.8}$ | $88.0_{\pm05.6}$ | $78.3_{\pm08.2}$ |
| BaRISTA | $89.3_{\pm05.0}$ | $81.2_{\pm07.2}$ | $89.0_{\pm05.7}$ | $80.9_{\pm07.2}$ |
| **RECTOR** | $\mathbf{92.1^*_{\pm04.1}}$ | $\mathbf{84.7^*_{\pm06.4}}$ | $\mathbf{92.4^*_{\pm03.9}}$ | $\mathbf{84.2^*_{\pm06.6}}$ |

### 3.2. Classification Results

We pre-trained RECTOR and all self-supervised baselines exclusively on each target dataset. Table 1 presents classification results on SEED, SEED-IV, and DEAP. RECTOR

*Table 3.* Ablation studies on SEED, SEED-IV, DEAP, MSIT, and ECR (w-F1: Weighted F1 (%); BAcc: Balanced Accuracy (%); AUROC (%)). Performance is averaged over subject-dependent and subject-independent settings for SEED, SEED-IV, and DEAP. **BOLD** indicates the best result. * denotes it is significantly worse than the full model.

| Model | SEED | | SEED-IV | | DEAP-Valence | | DEAP-Arousal | | MSIT | | ECR | |
|---|---|---|---|---|---|---|---|---|---|---|---|---|
| | w-F1 | BAcc | w-F1 | BAcc | w-F1 | BAcc | w-F1 | BAcc | AUROC | BAcc | AUROC | BAcc |
| $-\mathcal{L}_{\text{MPM:input}}$ | $71.6^*_{\pm08.8}$ | $71.6^*_{\pm08.6}$ | $52.4^*_{\pm08.4}$ | $51.4^*_{\pm08.2}$ | $66.5^*_{\pm10.9}$ | $65.2^*_{\pm10.7}$ | $67.9^*_{\pm12.8}$ | $65.7^*_{\pm12.9}$ | $90.3^*_{\pm05.1}$ | $82.8^*_{\pm06.9}$ | $90.2^*_{\pm04.8}$ | $82.3^*_{\pm06.8}$ |
| $-\mathcal{L}_{\text{MPM:rep}}$ | $71.4^*_{\pm08.9}$ | $71.5^*_{\pm09.1}$ | $52.4^*_{\pm08.5}$ | $51.6^*_{\pm08.0}$ | $66.3^*_{\pm10.1}$ | $65.1^*_{\pm10.9}$ | $67.5^*_{\pm12.8}$ | $65.2^*_{\pm12.4}$ | $90.4^*_{\pm04.9}$ | $82.4^*_{\pm06.5}$ | $90.5^*_{\pm04.7}$ | $82.7^*_{\pm07.0}$ |
| $-\mathcal{L}_{\text{TSM}}$ | $71.1^*_{\pm08.4}$ | $70.9^*_{\pm08.8}$ | $52.0^*_{\pm08.4}$ | $51.2^*_{\pm08.3}$ | $66.3^*_{\pm11.0}$ | $64.9^*_{\pm10.5}$ | $67.5^*_{\pm13.2}$ | $65.4^*_{\pm12.7}$ | $89.5^*_{\pm04.8}$ | $81.8^*_{\pm06.7}$ | $89.6^*_{\pm05.0}$ | $81.4^*_{\pm06.9}$ |
| $-\mathcal{L}_{\text{CVC}}$ | $71.9^*_{\pm09.0}$ | $71.6^*_{\pm09.0}$ | $53.1^*_{\pm07.9}$ | $52.7^*_{\pm08.4}$ | $67.0^*_{\pm10.6}$ | $65.4^*_{\pm10.2}$ | $68.4^*_{\pm12.6}$ | $67.2^*_{\pm12.9}$ | $90.6^*_{\pm04.7}$ | $83.1^*_{\pm06.5}$ | $91.1^*_{\pm04.4}$ | $83.5^*_{\pm06.8}$ |
| $-$AFP | $71.6^*_{\pm09.2}$ | $71.6^*_{\pm09.3}$ | $53.3^*_{\pm08.5}$ | $52.2^*_{\pm08.2}$ | $67.5^*_{\pm10.7}$ | $65.6^*_{\pm10.9}$ | $68.1^*_{\pm12.9}$ | $66.3^*_{\pm13.2}$ | $90.2^*_{\pm04.7}$ | $82.6^*_{\pm06.7}$ | $90.2^*_{\pm04.7}$ | $82.9^*_{\pm07.2}$ |
| $-$top-$p$ | $72.1^*_{\pm09.2}$ | $71.7^*_{\pm08.8}$ | $53.4^*_{\pm07.9}$ | $52.2^*_{\pm08.2}$ | $67.4^*_{\pm10.5}$ | $65.9^*_{\pm10.0}$ | $68.6^*_{\pm12.8}$ | $66.9^*_{\pm12.5}$ | $91.0^*_{\pm04.7}$ | $83.4^*_{\pm06.5}$ | $91.7^*_{\pm04.4}$ | $83.9^*_{\pm06.9}$ |
| $-\mathcal{L}_{\text{MTRL}}$ | $70.3^*_{\pm09.6}$ | $70.1^*_{\pm09.4}$ | $51.1^*_{\pm08.5}$ | $50.2^*_{\pm08.8}$ | $65.5^*_{\pm10.7}$ | $64.5^*_{\pm10.4}$ | $66.7^*_{\pm12.9}$ | $65.3^*_{\pm12.7}$ | $89.3^*_{\pm04.0}$ | $81.6^*_{\pm06.4}$ | $89.1^*_{\pm04.9}$ | $81.7^*_{\pm07.6}$ |
| Full SA | $70.2^*_{\pm09.2}$ | $69.8^*_{\pm09.0}$ | $52.3^*_{\pm08.9}$ | $51.1^*_{\pm09.2}$ | $65.1^*_{\pm11.2}$ | $64.2^*_{\pm10.9}$ | $66.0^*_{\pm12.6}$ | $64.8^*_{\pm13.5}$ | $89.0^*_{\pm04.8}$ | $81.4^*_{\pm06.7}$ | $89.2^*_{\pm04.7}$ | $81.5^*_{\pm07.0}$ |
| **RECTOR** | $\mathbf{73.1}_{\pm08.5}$ | $\mathbf{73.0}_{\pm08.7}$ | $\mathbf{54.2}_{\pm07.9}$ | $\mathbf{53.1}_{\pm08.0}$ | $\mathbf{68.1}_{\pm10.2}$ | $\mathbf{66.8}_{\pm10.6}$ | $\mathbf{69.0}_{\pm12.4}$ | $\mathbf{67.7}_{\pm12.6}$ | $\mathbf{92.1}_{\pm04.1}$ | $\mathbf{84.7}_{\pm06.4}$ | $\mathbf{92.4}_{\pm03.9}$ | $\mathbf{84.2}_{\pm06.6}$ |

establishes a new state-of-the-art across all settings, consistently outperforming the strongest task-specific baselines (MMM, mdJPT) and general approaches, including foundation models (PopT, CBraMod, CSBrain) by > 3% on average (Appendix C.1 also shows RECTOR achieves the best performance using Cohen's kappa). Table 2 shows that RECTOR outperforms all sEEG/iEEG baselines on MSIT and ECR. Moreover, RECTOR exhibits the lowest standard deviations in both tasks, underscoring its superior robustness over other baselines. Beyond that, RECTOR demonstrates superior training efficiency in both SSL and supervised fine-tuning compared to SSL methods that use full spatio-temporal self-attention (Fig. 9).

## 3.3. Ablation Studies

We conducted ablation studies to validate the individual contributions of RECTOR's core components (Table 3). We systematically removed modules targeting specific challenges identified in Section 1: (1) Masked Predictive Modeling components: input reconstruction ($-\mathcal{L}_{\text{MPM:input}}$) and representation alignment ($-\mathcal{L}_{\text{MPM:rep}}$); (2) Topological Structure Modeling ($-\mathcal{L}_{\text{TSM}}$); and (3) Cross-View Consistency ($-\mathcal{L}_{\text{CVC}}$). Additionally, we evaluated removing the entire Masked Topology and Representation Learning framework ($-\mathcal{L}_{\text{MTRL}}$), replacing it with standard MAE-style random masking. For the architecture, we ablated Adaptive Functional Partitioning ($-$AFP) and top-$p$ gating. Finally, Full SA replaces RECTOR-SA with dense spatio-temporal self-attention (also removing AFP and $\mathcal{L}_{\text{TSM}}$).

Table 3 confirms that every component is essential for optimal performance. Ablating the individual loss terms reveals that both $\mathcal{L}_{\text{MPM:input}}$ and $\mathcal{L}_{\text{MPM:rep}}$ are critical, with their removal leading to consistent performance drops across all benchmarks. Furthermore, removing $\mathcal{L}_{\text{TSM}}$ causes a sharp decline, validating the necessity of explicitly modeling region-channel assignments. Similarly, the removal of AFP consistently degrades results across all datasets, confirming that learned functional regions capture neural dynamics better than fixed anatomical priors. Crucially, the Full SA baseline performs significantly worse than RECTOR despite its higher computational cost, demonstrating

that RECTOR's block-sparse attention is not only more efficient but also learns more robust representations by filtering irrelevant connections. Finally, the largest drop observed in $-\mathcal{L}_{\text{MTRL}}$ highlights that our unified multi-view objective is far superior to simple reconstruction-based pre-training.

## 3.4. Scalability and Topological Generalization

Fig. 5 reports downstream classification under progressively larger pre-training regimes (Appendix F.8 for implementation details), ranging from (1) training from scratch (RECTOR-TFS) to pre-training on (2) each individual dataset (RECTOR), (3) all candidate datasets (**RECTOR+**, CHB-MIT (Shoeb, 2009) is added as a large candidate for EEG), (4) all candidate datasets using a larger RECTOR with more parameters (**RECTOR-L+**). The results demonstrate a clear scaling trend: performance consistently improves with both the amount of pre-training data and the model's size, with RECTOR-L+ consistently achieving the best results.

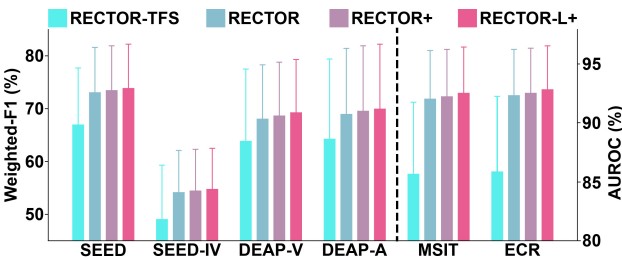

*Figure 5.* **Benefit of Expanded Pre-training.** Performance is averaged over subject-dependent and subject-independent settings for SEED, SEED-IV, and DEAP.

We further evaluate RECTOR's resilience to topological shifts (Fig. 6). To simulate real-world sensor failure, we evaluated zero-shot performance under random channel dropout rates ranging from 0% to 50%. As shown in Fig. 6(a), RECTOR exhibits superior stability compared to state-of-the-art baselines (e.g., mdJPT, BaRISTA). The divergence between RECTOR and the baselines at high missing rates confirms that RECTOR maintains a significantly flatter decay, effectively reconstructing missing local features from the global context. We also tested generalization to unseen sensor

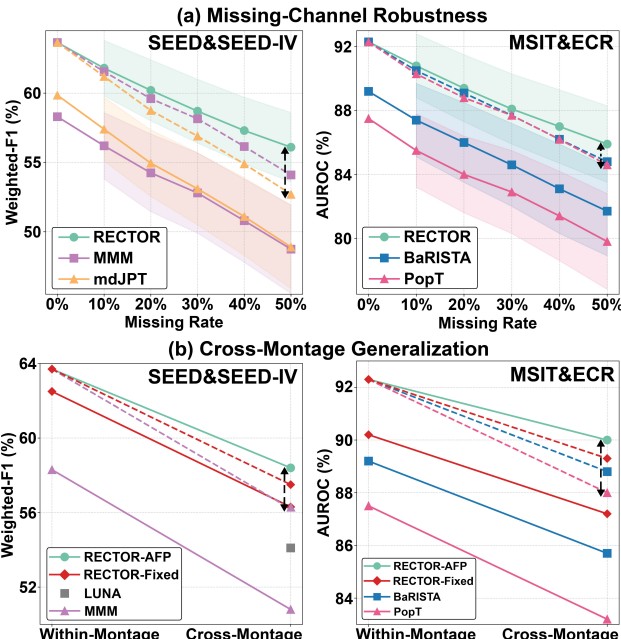

*Figure 6.* **Topological Robustness Analysis.** Baseline curves are vertically aligned (dashed lines) to match RECTOR's initial performance to compare the relative decay. SEED/SEED-IV performance is averaged over SEED and SEED-IV, and over subject-dependent and subject-independent settings.

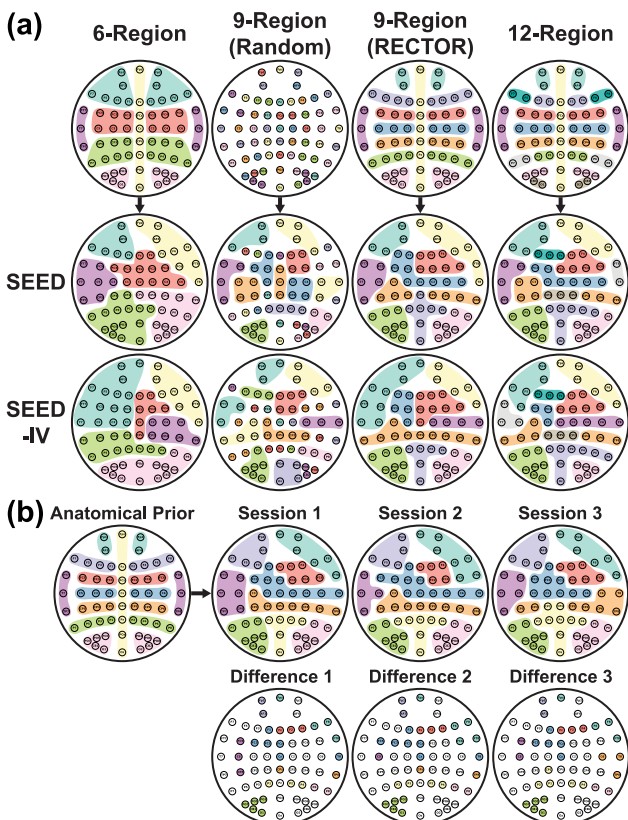

*Figure 7.* **AFP reveals structured refinements beyond fixed anatomy.** (a) Anatomical priors and learned AFP partitions under different region granularities on SEED and SEED-IV. (b) Anatomical prior versus session-wise learned AFP consensus partitions on SEED, with the bottom row highlighting reassigned channels.

layouts (Fig. 6(b)), specifically transferring representations from DEAP to SEED/SEED-IV and performing mutual transfer between MSIT and ECR. While fixed-topology models suffer significant performance drops when transferring across montages, RECTOR-AFP minimizes this decay. The marked gap between RECTOR-AFP and RECTOR-Fixed demonstrates that learning content-aware region tokens yields a representation that is structurally invariant to physical montage shifts, unlike rigid anatomical definitions.

### 3.5. Interpretability Analysis

Fig. 7 compares AFP-learned partitions with the anatomical prior across representative region granularities and sessions. As shown in Fig. 7(a), AFP does not simply preserve the initialization: across different region granularities, the learned partitions remain spatially contiguous while progressively refining the same macro-scale organization, with a smaller number of regions $R$ producing coarser regions and a larger $R$ yielding finer subdivisions. This trend is robust to moderate changes in granularity, while the random 9-region initialization leads to a visibly less coherent organization and inferior performance than the default anatomy-initialized setting (e.g., SEED SD w-F1: 80.9 vs. 85.0, see Appendix C.5), indicating that AFP depends on meaningful partition structure rather than region count alone.

Fig. 7(b) further shows that AFP learns nontrivial yet spatially coherent refinements beyond the anatomical prior.

Starting from the anatomical initialization, AFP consistently reshapes the topology into smoother data-driven groupings by merging nearby channels that were previously separated and splitting anatomically broad regions into more refined substructures. These refinements are reproducible: AFP reassigns 36.5–51.5% of channels from anatomy across datasets, and the reassigned channels remain highly consistent across sessions on SEED and SEED-IV (84.7%/86.1%, see Appendix C.6), supporting that AFP learns stable functional refinements rather than arbitrary rearrangements. Beyond this, the learned partitions repeatedly form an anterior subgroup (Fz/F2/F4), separate nearby fronto-central electrodes (F1/FC3/FC1/FCz/Cz), regroup the left lateral chain (FC5/C5/CP5) with left temporal electrodes, and induce a more asymmetric posterior parietal-occipital organization. Taken together, these results support interpreting AFP-learned regions as task-adaptive grouping variables for structured attention, rather than as arbitrary internal clusters or definitive neuroscientific parcellations. These analyses provide supportive evidence that the learned topology is structured and neurophysiologically plausible, while external neuroscientific validation remains beyond the scope of this work.

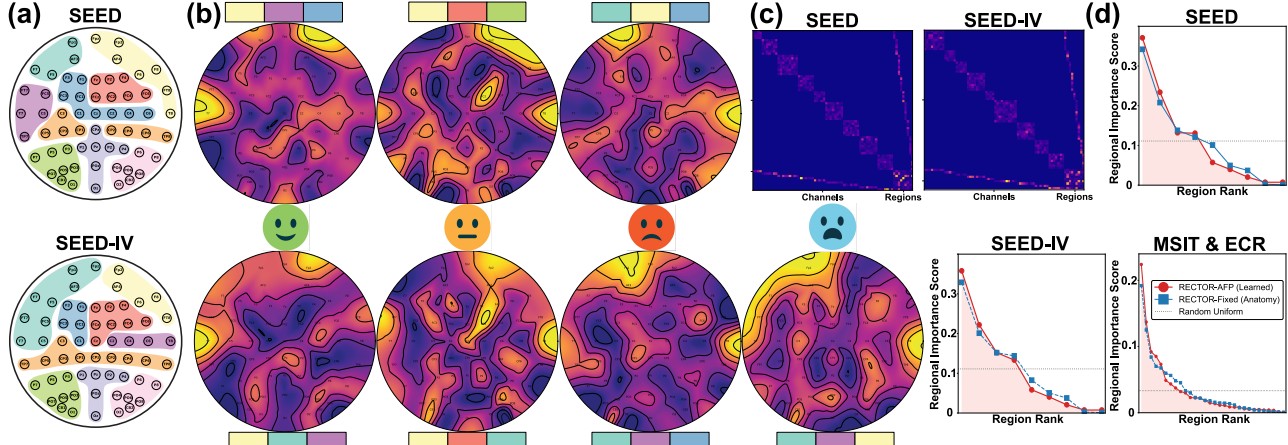

*Figure 8.* **Interpretability of RECTOR.** (a) AFP-learned region-channel partitioning for SEED and SEED-IV. (b) Channel-level class activation maps on SEED (top) and SEED-IV (bottom). Bars color-coded as in (a) indicate the top-3 contributing regions. (c) Region-Channel attention matrices reveal a sparse, block-diagonal structure. (d) Distribution of regional importance. RECTOR-AFP exhibits a sharper decay than fixed-anatomy, confirming that the learned topology effectively isolates discriminative functional regions.

Fig. 8(a) and (b) illustrate learned region-channel assignments and the corresponding class activation maps. For the channel-level maps in Fig. 8(b), we use Grad-CAM (Selvaraju et al., 2017): after forwarding an input trial through the fine-tuned model, we apply Grad-CAM to the output layer before the classification head to obtain a per-channel importance score for the predicted class, and then average the resulting maps across subjects within each dataset (see Appendix F.3). The model identifies distinct spatial signatures for different emotional states. Consistent with the motivational-direction models of frontal asymmetry (Harmon-Jones, 2003; 2004; Light et al., 2009; Parro et al., 2018), we observe opposing patterns of frontal lobe engagement for positive versus negative emotions in both SEED and SEED-IV datasets. The color-coded bars in Fig. 8(b) further highlight that the top-3 contributing regions consistently align with fronto-temporal and occipito-temporal areas, known hubs for emotional processing (Sun et al., 2023).

The attention matrices in Fig. 8(c) reveal a block-diagonal structure, and the blocks representing attention between regions and between regions-channels are more visibly active, indicating that RECTOR-SA successfully transfers the modeling of long-range functional connectivity away from the noisy channel space and concentrates it in the high-signal region space. This is further summarized by the Regional Importance Score (RIS) distribution in Fig. 8(d). RIS is a normalized region-level attribution score whose values sum to 1, so a sharper decay indicates that the attribution mass is concentrated in fewer top-ranked regions. Compared to the fixed anatomical baseline RECTOR-Fixed, RECTOR-AFP exhibits a sharper decay in RIS. This pronounced sparsity confirms that the learned topology effectively suppresses background noise and isolates the most discriminative func-

tional regions for downstream predictions.

## 4. Conclusion

We introduced RECTOR, the first end-to-end, self-supervised framework that unifies region, channel, and temporal modeling for neural representation learning beyond fixed anatomical priors. By leveraging RECTOR-SA, a hierarchical, block-sparse self-attention induced by Adaptive Functional Partitioning, our model evolves static anatomical definitions into adaptive functional regions essential for capturing distributed brain dynamics. The Masked Topology and Representation Learning strategy synergistically optimizes predictive modeling, topological structure, and cross-view consistency, establishing a new state-of-the-art in EEG emotion recognition and sEEG task-engagement classification. Furthermore, our results demonstrate RECTOR's superior robustness to missing channels and cross-montage generalization, underscoring its potential for large-scale pre-training on heterogeneous datasets. Finally, our analyses show that AFP learns stable functional refinements beyond fixed anatomy, while the resulting region- and channel-level visualizations support neurophysiologically plausible representations. These properties pave the way for more interpretable neurocognitive diagnostics and adaptive, personalized neuro-interventions.

## Acknowledgements

This work was supported in part by the National Institute of Mental Health under Grant R01-MH-123634 and in part by the Swiss State Secretariat for Education, Research and Innovation under Contract SCR0548363. We gratefully acknowledge Dr. Alik Widge for providing access to the sEEG data used in this study.

## Impact Statement

RECTOR pushes the frontier of self-supervised neural representation learning, offering a scalable path toward objective, data-driven biomarkers for affective and cognitive disorders. By introducing Adaptive Functional Partitioning (AFP) and Masked Topology and Representation Learning (MTRL), RECTOR moves beyond rigid anatomical priors to capture the adaptive, distributed nature of brain network dynamics. This capability significantly improves the precision of both non-invasive (EEG) and invasive (sEEG) diagnostics, enabling earlier detection and more personalized interventions. Crucially, RECTOR's demonstrated zero-shot robustness to sensor failure and cross-montage generalization addresses two of the most persistent barriers to clinical deployment: data heterogeneity and hardware variability. This positions RECTOR as a viable foundation for large-scale pre-training across diverse, incompatible datasets. Furthermore, its interpretable region-channel attention maps provide clinicians with transparent, neurophysiologically plausible insights into functional brain activity, fostering trust and facilitating novel neuroscientific discovery.

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

# A. Comparisons with Related Works

Learning effective neural representations underlying cognitive conditions from EEG and sEEG signals is a central challenge in cognitive neuroscience. We review prior work in four key areas: CNN-based and hybrid models, graph neural networks, transformers for spatio-temporal modeling, and self-supervised, region-aware learning frameworks.

## A.1. CNN and Hybrid Architectures

Convolutional Neural Networks (CNNs) are widely used for their ability to extract local spatio-temporal features from EEG signals. Models like TSception (Ding et al., 2022) and MASA-TCN (Ding et al., 2024) employ multi-scale temporal convolutions to capture neural dynamics across different frequencies and durations. Hybrid architectures such as EEG Conformer (Song et al., 2022) and REmoNet (Jiang et al., 2024b) combine CNNs with transformers or RNNs to pair local feature extraction with long-range dependency modeling. While effective, these methods often rely on fixed convolutional filters for spatial processing and can struggle to flexibly model the brain's dynamic, long-range functional interactions. In contrast, RECTOR uses a fully attention-based approach (RECTOR-SA) that learns these relationships adaptively from data, guided by a flexible, anatomically-informed structure rather than rigid filters.

## A.2. Graph Neural Networks for Spatial Modeling

Graph Neural Networks (GNNs) have been proposed to explicitly model the brain's network topology. LGGNet (Ding et al., 2023) and PGCN (Jin et al., 2024) represent electrodes as nodes in a graph, using graph convolutions to learn spatial features that respect neurophysiological priors. These models often use a hierarchical or pyramidal structure to aggregate information from local electrodes to global brain regions. GNN-based approaches typically rely on a pre-defined, static graph of electrode connectivity. RECTOR differs by introducing Adaptive Functional Partitioning (AFP). AFP learns to dynamically partition channels into task-specific functional regions, allowing the network topology to adapt to individual physiological variations and changing cognitive states rather than relying on a fixed anatomical adjacency matrix.

## A.3. Transformers for Spatio-Temporal Modeling

Transformers are the dominant architecture for sequence modeling, but adapting them for the joint spatio-temporal nature of EEG is non-trivial. Prior works fall into several categories:

1. Decoupled Attention: Many models handle spatial and temporal dimensions separately to maintain efficiency. CBraMod (Wang et al., 2025) uses criss-cross attention to model channel-wise and time-wise dependencies in parallel. Others like Brant and its successor Brant-2 (Yuan et al., 2024b), BrainWave (Yuan et al., 2024a), Seegnificant (Mentzelopoulos et al., 2024), and EmT (Ding et al., 2025) apply sequential transformer blocks. The primary limitation of these methods is their failure to capture joint, simultaneous spatio-temporal interactions.

2. Full Spatio-Temporal Attention: Foundation models like LaBraM (Jiang et al., 2024a) apply a dense, full self-attention across all channel-time tokens. While this enables the modeling of all possible interactions, it is computationally prohibitive and risks amplifying noise.

3. Generalist Architectures: Some works adapt general-purpose time-series transformers like Medformer for medical data. While powerful, these generic models lack the specific inductive biases for neurophysiological signals.

RECTOR addresses these limitations via RECTOR-SA. Driven by AFP, it models joint region-channel-temporal interactions through a hierarchical, block-sparse mechanism with top-$p$ gating. This design strikes an optimal balance: it captures essential interactions within learned functional hubs while efficiently attending to global region-level dynamics, avoiding the computational cost of full attention and the expressivity bottlenecks of decoupled approaches. Unlike generalist models, RECTOR is grounded in adaptive neurophysiology, evolving region definitions from anatomical priors.

## A.4. Self-Supervised and Region-Aware Learning

SSL is critical for leveraging sparsely labeled EEG/sEEG data. While many works use masked modeling, some have begun to incorporate regional brain structure. MMM (Yi et al., 2023) introduces region-wise tokens, and Du-IN (Zheng et al., 2024) forms patch embeddings by fusing channels within specific regions. These approaches tackle important, often orthogonal,

challenges but do not offer a unified solution. PopT (Chau et al.) and Seegnificent (Mentzelopoulos et al., 2024) oversimplify brain topology by performing population-level spatial encoding, treating all channels as a single global region and thus ignoring the brain's established functional network architecture. While EEGPT (Wang et al., 2024) introduces a powerful dual objective with representation alignment, it lacks the explicit region-channel structure and dedicated regularization to learn a non-redundant, hierarchical representation. Similarly, Du-IN fuses channels, discarding the fine-grained channel-specific dynamics that RECTOR preserves and models. RECTOR distinguishes itself with a holistic Masked Topology and Representation Learning (MTRL) framework. Unlike standard masked modeling, MTRL synergizes three complementary objectives: Masked Predictive Modeling (MPM) for local feature reconstruction, Topological Structure Modeling (TSM) to explicitly regularize learned region-channel assignments, and Cross-View Consistency (CVC) to enforce invariance across structural views. This enables RECTOR to learn a robust hierarchy that preserves both fine-grained channel dynamics and high-level regional semantics.

## B. Methodology

### B.1. RECTOR-SA Computation and Complexity

Let $R$ be the number of regions, $T$ the number of time segments, and $d$ the embedding dimension. For simplicity, we assume a constant number of channels $P$ per region. The full spatio-temporal SA has a complexity of: (1) $\mathcal{O}(3RPTd^2)$ for $\mathbf{QKV}$ projections, (2) $\mathcal{O}(2R^2P^2T^2d)$ for $\mathbf{Q}$-$\mathbf{K}$ multiplication and $\mathbf{A}$-$\mathbf{V}$ multiplication, where $\mathbf{A} = \mathrm{softmax}(\frac{\mathbf{QK}^\top}{\sqrt{d}} \odot \mathbf{M}_{\mathrm{Attn}})$, (3) $\mathcal{O}(R^2P^2T^2)$ for softmax$(\cdot)$, and (4) $\mathcal{O}(RPTd^2)$ for output projection. Therefore, the total complexity of full self-attention is $\mathcal{O}(2R^2P^2T^2d + 4RPTd^2)$ for $\mathbf{QKV}$ multiplications and projections, and $\mathcal{O}(R^2P^2T^2)$ for softmax$(\cdot)$.

For RECTOR-SA, we can split it into:

1. Local functional attention: We split the $RP$ channels into $R$ regions and compute self-attention within each region over the full temporal extent $T$. The complexity of its $\mathbf{QKV}$ multiplications is $\mathcal{O}(2R(PT)^2d)$. We compute softmax$(\cdot)$ on a matrix of size $P \times (PT + 1)$ for $RT$ times. Therefore, the complexity of softmax$(\cdot)$ is $\mathcal{O}(RTP(PT + 1)) \approx \mathcal{O}(RP^2T^2)$.

2. Global functional attention: We treat region tokens plus conditioning tokens as a sequence of length $RT + K$. The complexity of its $\mathbf{QKV}$ multiplications is $\mathcal{O}(2(RT + K)^2d) \approx \mathcal{O}(2R^2T^2d)$. We compute softmax$(\cdot)$ on a matrix of size $(RT + k) \times (RPT + k)$ attention block. Therefore, the complexity of softmax$(\cdot)$ is $\mathcal{O}((RT + K)(RPT + k)) \approx \mathcal{O}(R^2PT^2)$.

3. Anatomical attention: We allow each channel token to only attend to its corresponding region token at the same time index and vice versa. The complexity of its $\mathbf{QKV}$ multiplications is $\mathcal{O}(2RPTd)$, which is a lower order term and can be ignored.

4. Total projection: The complexity of all projections is $\mathcal{O}(4(RPT + RT + k)d^2) \approx \mathcal{O}(4RPTd^2 + 4RTd^2)$.

Therefore, the complexity of $\mathbf{QKV}$ multiplications and projections in RECTOR-SA is $\mathcal{O}(2RP^2T^2d + 2R^2T^2d + 4RPTd^2 + 4RTd^2)$, and the complexity for softmax$(\cdot)$ is $\mathcal{O}(RP^2T^2 + R^2PT^2)$. For $\mathbf{QKV}$ operations, RECTOR-SA is more efficient when $d < T(P^2(R - 1) - R)/2$. For the softmax$(\cdot)$ computation, RECTOR-SA is advantageous when $P > R/(R - 1)$.

Additionally, unlike approaches that require separate forward passes for each prediction target, RECTOR computes all five masked-region predictions in a single forward pass.

### B.2. Region-Channel Regularization (RCReg)

We further augment our SSL with **RCReg**: Region-Channel Regularization. This module provides dedicated guidance for region and channel learning using a variance hinge loss $\mathcal{L}_{\mathrm{RCVar}}(\cdot)$ and a covariance regularization $\mathcal{L}_{\mathrm{RCCov}}(\cdot)$ on representations $\mathbf{Z}$.

Let $B$ be the batch size, $N$ the number of tokens, and $d$ the embedding size. Given a batch of hidden representations $\mathbf{Z} = \mathbf{Z_c}\mathbf{W}_{\mathrm{Reg}} \in \mathbb{R}^{B \times N \times d}$, extracted by the context encoder $\mathbf{Z_c}$ and followed by a linear projection $\mathbf{W}_{\mathrm{Reg}}$, we can define RCReg as a combination of a variance hinge loss and a covariance loss on region-channel tokens $\mathcal{L}_{\mathrm{RCReg}} =$

$\lambda_{\text{RCVar}} \mathcal{L}_{\text{RCVar}}(\mathbf{Z}) + \lambda_{\text{RCCov}} \mathcal{L}_{\text{RCCov}}(\mathbf{Z})$:

$$\mathcal{L}_{\text{RCVar}}(\mathbf{Z}) = \frac{1}{N^{\text{RC}}} \sum_{n=1}^{N^{\text{RC}}} \max \left( 0, \gamma - \sqrt{\frac{1}{BTd-1} \sum_{b=1}^{B} \sum_{t=1}^{T} \sum_{k=1}^{d} \left( \mathbf{Z}_{b,t,k}^{(n)} - \bar{\mathbf{Z}}^{(n)} \right)^2 + \epsilon} \right)$$

$$\mathcal{L}_{\text{RCCov}}(\mathbf{Z}) = \frac{1}{N^{\text{RC}}} \sum_{i \neq j} \left[ \frac{1}{BTd-1} \sum_{b=1}^{B} \sum_{t=1}^{T} \sum_{k=1}^{d} (\mathbf{Z}_{b,t,k} - \bar{\mathbf{Z}})(\mathbf{Z}_{b,t,k} - \bar{\mathbf{Z}})^T \right]_{i,j}^2$$

(13)

Here, $N^{\text{RC}}$ is the number of region and channel tokens. Sample means $\bar{\mathbf{Z}}$ are computed across specified sample axes.

**Motivation**   RCReg operates on the token dimension to solve a new problem: hierarchical representational mixing. Its two components serve distinct purposes: $\mathcal{L}_{\text{RCVar}}(\mathbf{Z})$ prevents collapse in the representations by ensuring that the variance of each region/channel token's embeddings stays above a threshold. $\mathcal{L}_{\text{RCCov}}(\mathbf{Z})$ encourages region/channel token decorrelation in two conditions: (1) It minimizes the correlation between channel tokens from the same region. (2) It minimizes the correlation between each channel token and its corresponding region token. Therefore, RCReg compels the model to propagate shared, regional information exclusively into the region token. This avoids trivial region features and explicitly encourages region and channel tokens to encode *region-common* and *channel-specific* information.

It should be noted that when using RCReg, we should treat $\lambda_{\text{RCVar}}$ and separate $\lambda_{\text{RCCov}}$ on $\mathbf{C}^{\text{RR}}$, $\mathbf{C}^{\text{RC}}$, $\mathbf{C}^{\text{CR}}$, and $\mathbf{C}^{\text{CC}}$ as hyperparameters during self-supervised learning.

## B.3. Input Reconstruction Loss Mitigates Representation Collapse

Here is a detailed theoretical analysis grounded in the mutual information maximization to justify why our $\text{NC}^2$-MM loss mitigates representation collapse and yields superior representations.

We want the learned representation $\mathbf{Z}$ to capture the maximum amount of information about the input $\mathbf{X}$. That is, we want to maximize the mutual information $I(\mathbf{X}; \mathbf{Z})$, where:

$$I(\mathbf{X}; \mathbf{Z}) = H(\mathbf{X}) - H(\mathbf{X}|\mathbf{Z})$$

(14)

Since the entropy of the dataset $H(\mathbf{X})$ is constant, maximizing $I(\mathbf{X}; \mathbf{Z})$ is equivalent to minimizing the conditional entropy $H(\mathbf{X}|\mathbf{Z})$. If we model the conditional distribution $P(\mathbf{X}|\mathbf{Z})$ as a Gaussian distribution $\mathcal{N}(\hat{\mathbf{X}}(\mathbf{Z}), \sigma^2 \mathbf{I})$, then minimizing the negative log-likelihood corresponds exactly to minimizing the Mean Squared Error (MSE):

$$\mathcal{L}_{\text{input}} = ||\mathbf{X} - \hat{\mathbf{X}}(\mathbf{Z})||^2 \propto -\log P(\mathbf{X}|\mathbf{Z})$$

(15)

The expectation of this negative log-likelihood over the data distribution is the conditional entropy (plus a constant):

$$\mathbb{E}_{\mathbf{X}, \mathbf{z}}[-\log P(\mathbf{X}|\mathbf{Z})] = H(\mathbf{X}|\mathbf{Z}) + C$$

(16)

Therefore, minimizing $\mathcal{L}_{\text{input}}$ over the data distribution minimizes the conditional entropy $H(\mathbf{X}|\mathbf{Z})$.

A collapsed representation where $\mathbf{Z} = c$ (constant) has $I(\mathbf{X}, \mathbf{Z} = c) = 0$ and maximizes conditional entropy : $H(\mathbf{X}|\mathbf{Z} = c) = H(\mathbf{X})$. By minimizing $\mathcal{L}_{\text{input}}$, we force $H(\mathbf{X}|\mathbf{Z})$ to be lower than $H(\mathbf{X})$, which guarantees that $I(\mathbf{X}; \mathbf{Z}) > 0$. It shows that the global minimum of $\mathcal{L}_{\text{input}}$ cannot be a collapsed state. The reconstruction loss forces $\mathbf{Z}$ to retain information about $\mathbf{X}$.

The effect of $\mathcal{L}_{\text{input}}$ can also be understood via variance analysis. Consider the loss function $\mathcal{L}_{\text{input}} = \mathbb{E}[||\mathbf{X} - \hat{X}(\mathbf{Z})||^2]$. If we assume the encoder collapses such that $\mathbf{Z} = c$ for all $\mathbf{X}$. The decoder $\hat{\mathbf{X}}(\mathbf{Z})$ then outputs a constant vector $\mu = \hat{\mathbf{X}}(c)$. The optimal constant vector $\mu$ that minimizes MSE is the mean of the dataset: $\mu = \mathbb{E}[\mathbf{X}]$. In this collapsed state, the loss becomes the variance of the dataset:

$$\mathcal{L}_{\text{input}}^{\text{collapse}} = \mathbb{E}[||\mathbf{X} - \mathbb{E}[\mathbf{X}]||^2] = \text{Var}(\mathbf{X})$$

(17)

For any non-trivial dataset, $\text{Var}(\mathbf{X}) > 0$. A non-collapsed encoder that retains even a single bit of information about $\mathbf{X}$ can achieve a lower reconstruction error than $\text{Var}(\mathbf{X})$. The collapsed state $\mathbf{Z} = c$ is never the global minimum of $\mathcal{L}_{\text{input}}$. The optimization landscape of $\mathcal{L}_{\text{input}}$ inherently drives the model away from collapse.

### B.4. Top-$p$ Gating

Given an input vector $\mathbf{x} \in \mathbb{R}^n$ with decreasing order $\{x_1, \ldots, x_n\}$, we first compute its softmax-normalized probabilities $\mathbf{p} = \text{softmax}(\mathbf{x}) = \{p_1, \ldots, p_n\}$. For a fixed threshold $p \in (0, 1]$, we define the top-$p$ gating index $\tau$ as the smallest integer satisfying:

$$\sum_{i=1}^{\tau} p_i \geq p \tag{18}$$

The resulting **top-$p$ gating mask** (Figure 3) $\text{top-}p(\mathbf{x}) = \mathbf{m} \in \{0, 1\}^n$ is then:

$$m_i = \begin{cases} 1, & i \leq \tau \\ 0, & i > \tau \end{cases} \tag{19}$$

and the gated vector $\mathbf{x}_g$ is obtained by elementwise multiplication:

$$\mathbf{x}_g = \mathbf{x} \odot \text{top-}p(\mathbf{x}) \tag{20}$$

Integrated into RECTOR-SA (see Section 2.3), this top-$p$ gating mechanism enables the model to dynamically focus on the most informative region-channel-temporal attention entries–automatically adjusting sparsity to the data's distribution–without the need to hand-tune a fixed amount of entries in top-$k$ gating (Shazeer et al., 2017), yielding both great adaptivity and computational efficiency in RECTOR-SA.

### B.5. RECTOR-Transformer Block

The **RECTOR-Transformer block** serves as the core building block in our encoder, predictor, and decoder. It comprises two key components: (1) Multi-head RECTOR-SA (see Section 2.3), which performs both structured and dynamic token mixing across spatial (region, channel) and temporal dimensions (Figure 3); (2) SwiGLU-style feed-forward network (Shazeer, 2020), which further enhances feature learning.

Concretely, given the post-attention tokens $\mathbf{X}$, we apply:

$$\begin{aligned} \mathbf{X}_{\text{FFN}} &= \text{FFN}_{\text{SwiGLU}}(\mathbf{X}) \\ &= (\text{Swish}_1(\mathbf{X}\mathbf{W}_1) \odot \mathbf{X}\mathbf{V})\mathbf{W}_2 \end{aligned} \tag{21}$$

where $\text{Swish}_\beta(x) = x\sigma(\beta x)$. With residual connections and layer normalization, the $\ell_{\text{th}}$ block is:

$$\begin{aligned} \mathbf{X}_{\text{SA}}^{(\ell)} &= \text{RECTOR-SA}(\text{Norm}(\mathbf{X}^{(\ell)})) + \mathbf{X}^{(\ell)} \\ \mathbf{X}^{(\ell+1)} &= \text{FFN}_{\text{SwiGLU}}(\text{Norm}(\mathbf{X}_{\text{SA}}^{(\ell)})) + \mathbf{X}_{\text{SA}}^{(\ell)} \end{aligned} \tag{22}$$

Here, $\mathbf{X}^{(\ell)}$ and $\mathbf{X}^{(\ell+1)}$ denote the input and output token matrices of the $\ell_{\text{th}}$ block.

## C. Results

### C.1. Classification Results

We also report Cohen's kappa in Table 4 on SEED, SEED-IV, and DEAP. Across all datasets, RECTOR maintains the top performance using Cohen's kappa, and it significantly outperforms the other baselines in all comparisons.

We have also evaluated all models on MSIT and ECR using 5-fold cross-validation. Results are reported in Table 5, demonstrating a consistent performance with our chronological split and confirming that our findings are robust under both evaluation protocols.

*Table 4.* Classification performance on SEED, SEED-IV, and DEAP (Cohen's Kappa (%)). SD: Subject Dependent; SI: Subject Independent. **BOLD** and UNDERLINE indicate the best and second-best results. * denotes it significantly outperforms the second-best model.

| Model | SEED | | SEED-IV | | DEAP-Valence | | DEAP-Arousal | |
|---|---|---|---|---|---|---|---|---|
| | SD | SI | SD | SI | SD | SI | SD | SI |
| TSception | $56.99_{\pm 15.77}$ | $23.60_{\pm 10.50}$ | $33.90_{\pm 14.19}$ | $12.06_{\pm 05.12}$ | $20.47_{\pm 09.42}$ | $17.90_{\pm 08.61}$ | $21.74_{\pm 09.55}$ | $18.87_{\pm 09.11}$ |
| Conformer | $49.63_{\pm 17.30}$ | $25.43_{\pm 10.51}$ | $34.39_{\pm 13.48}$ | $14.01_{\pm 05.80}$ | $23.82_{\pm 09.22}$ | $20.50_{\pm 08.15}$ | $24.02_{\pm 09.92}$ | $20.15_{\pm 08.85}$ |
| LGGNet | $50.39_{\pm 16.30}$ | $22.75_{\pm 10.21}$ | $31.33_{\pm 13.15}$ | $13.01_{\pm 05.66}$ | $21.61_{\pm 09.24}$ | $17.41_{\pm 08.16}$ | $21.30_{\pm 09.92}$ | $21.42_{\pm 08.80}$ |
| PGCN | $60.54_{\pm 16.32}$ | $31.15_{\pm 09.59}$ | $39.24_{\pm 12.52}$ | $16.00_{\pm 05.89}$ | $24.79_{\pm 09.02}$ | $19.92_{\pm 08.01}$ | $23.58_{\pm 09.86}$ | $24.81_{\pm 08.83}$ |
| EmT | $59.49_{\pm 15.42}$ | $32.57_{\pm 09.79}$ | $36.54_{\pm 12.02}$ | $14.87_{\pm 06.17}$ | $23.78_{\pm 10.08}$ | $21.71_{\pm 09.05}$ | $23.63_{\pm 09.88}$ | $24.69_{\pm 09.38}$ |
| MMM | $61.68_{\pm 14.65}$ | $32.48_{\pm 09.23}$ | $40.32_{\pm 11.84}$ | $16.02_{\pm 05.20}$ | $24.31_{\pm 09.21}$ | $20.68_{\pm 08.81}$ | $22.55_{\pm 09.35}$ | $21.18_{\pm 09.61}$ |
| CBraMod | $64.48_{\pm 13.31}$ | $32.88_{\pm 09.45}$ | $39.81_{\pm 11.95}$ | $17.83_{\pm 05.61}$ | $24.83_{\pm 09.44}$ | $23.05_{\pm 08.29}$ | $26.48_{\pm 09.13}$ | $27.11_{\pm 09.52}$ |
| PopT | $62.69_{\pm 14.76}$ | $32.85_{\pm 09.35}$ | $37.25_{\pm 12.68}$ | $15.66_{\pm 05.81}$ | $26.73_{\pm 09.10}$ | $23.55_{\pm 08.96}$ | $25.88_{\pm 09.32}$ | $24.94_{\pm 08.60}$ |
| **RECTOR** | $\mathbf{71.87}^*_{\pm 09.23}$ | $\mathbf{37.56}^*_{\pm 10.12}$ | $\mathbf{45.52}^*_{\pm 10.12}$ | $\mathbf{22.30}^*_{\pm 05.51}$ | $\mathbf{33.51}^*_{\pm 09.45}$ | $\mathbf{27.59}^*_{\pm 08.42}$ | $\mathbf{34.34}^*_{\pm 09.43}$ | $\mathbf{32.07}^*_{\pm 09.38}$ |

*Table 5.* Classification performance on MSIT and ECR using 5-fold cross-validation (AUROC (%)). **BOLD**/UNDERLINE indicates the best/second-best results. * denotes it significantly outperforms the second-best model.

| Model | MSIT | ECR |
|---|---|---|
| Seegnificant | $84.08_{\pm 06.88}$ | $84.69_{\pm 07.05}$ |
| Brant | $86.42_{\pm 06.49}$ | $86.29_{\pm 06.27}$ |
| Du-IN | $86.68_{\pm 05.80}$ | $86.36_{\pm 05.94}$ |
| PopT | $86.58_{\pm 05.90}$ | $87.44_{\pm 05.62}$ |
| **RECTOR** | $\mathbf{90.32}^*_{\pm 05.42}$ | $\mathbf{90.39}^*_{\pm 05.57}$ |

## C.2. Training Efficiency

Across all benchmarks, RECTOR demonstrates superior training efficiency compared to self-supervised learning (SSL) methods that use full spatio-temporal self-attention (Fig. 9). In our analysis of training times for both SSL and supervised fine-tuning (SFT), RECTOR consistently outperformed MAE, JEPA, and a CAE baseline ($\times 1$). This efficiency gain is most evident on high-density neural recordings with many channels and regions (e.g., MSIT/ECR and SEED/SEED-IV), a direct result of our proposed RECTOR-SA architecture.

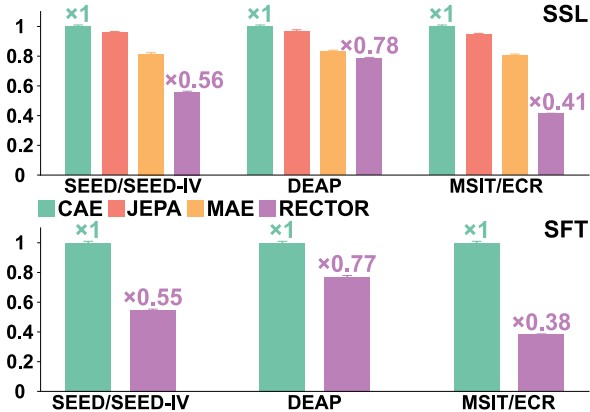

*Figure 9.* Training time comparison.

## C.3. Non-Contrastive Loss Analysis

In Table 6, we show the input reconstruction loss ($\mathcal{L}_{\text{MPM:input}}$) and representation alignment loss ($\mathcal{L}_{\text{MPM:rep}}$) of RECTOR and how the losses are changed if we remove the other loss component in self-supervised learning. Together with Table 3, we observed that removing $\mathcal{L}_{\text{MPM:input}}$ reduces $\mathcal{L}_{\text{MPM:rep}}$ but degrades downstream performance, indicating a slight feature collapse. Conversely, omitting $\mathcal{L}_{\text{MPM:rep}}$ causes a substantial increase in $\mathcal{L}_{\text{MPM:input}}$ and also worsens downstream results. These findings confirm that both objectives are essential.

*Table 6.* Ablation studies on MSE loss between predicted and ground-truth neural representations ($\mathcal{L}_{\text{MPM:rep}}$) and input signals ($\mathcal{L}_{\text{MPM:input}}$). We used standardized neural inputs. **BOLD** indicates the best result.

| Model | SEED | | SEED-IV | | DEAP | | MSIT | | ECR | |
|---|---|---|---|---|---|---|---|---|---|---|
| | $\mathcal{L}_{\text{MPM:rep}}$ | $\mathcal{L}_{\text{MPM:input}}$ | $\mathcal{L}_{\text{MPM:rep}}$ | $\mathcal{L}_{\text{MPM:input}}$ | $\mathcal{L}_{\text{MPM:rep}}$ | $\mathcal{L}_{\text{MPM:input}}$ | $\mathcal{L}_{\text{MPM:rep}}$ | $\mathcal{L}_{\text{MPM:input}}$ | $\mathcal{L}_{\text{MPM:rep}}$ | $\mathcal{L}_{\text{MPM:input}}$ |
| $-\mathcal{L}_{\text{MPM:rep}}$ | N/A | 0.3872 | N/A | 0.4165 | N/A | 0.4230 | N/A | 0.3828 | N/A | 0.3783 |
| $-\mathcal{L}_{\text{MPM:input}}$ | **0.0352** | N/A | **0.0197** | N/A | **0.0409** | N/A | **0.0261** | N/A | **0.0244** | N/A |
| **RECTOR** | 0.0413 | **0.2081** | 0.0200 | **0.1849** | 0.0457 | **0.2224** | 0.0293 | **0.2376** | 0.0264 | **0.2331** |

## C.4. Inter-Subject and Intra-Subject Variance

In Table 7, we provide a granular breakdown of the total standard deviation into inter-subject and intra-subject standard deviations for the SEED and SEED-IV subject-dependent (SD) protocols. The total standard deviation is calculated as the standard deviation across the weighted F1-Score of all subjects and sessions together. The inter-subject term is calculated as the standard deviation across the weighted F1-Score (averaged across sessions) of different subjects. The intra-subject term is calculated as the cross-subject average of the standard deviation (across sessions).

This rigorous analysis confirms our model's robustness: the majority of the total variance is attributed to differences between subjects, while the variability within the same subject across different sessions is relatively smaller. These results support that RECTOR learns stable representations for individual users over time.

*Table 7.* Total, inter-subject, and intra-subject standard deviation on SEED and SEED-IV under subject-dependent (SD) protocol (Weighted F1 Score (%)). **BOLD**: the best performance.

| Model | SEED:SD | | | SEED-IV:SD | | |
|---|---|---|---|---|---|---|
| | Total | Inter-Subject | Intra-Subject | Total | Inter-Subject | Intra-Subject |
| **RECTOR** | ±08.80 | ±06.82 | ±04.93 | ±11.16 | ±08.42 | ±06.63 |
| **RECTOR+** | **±08.70** | **±06.70** | ±05.04 | ±11.46 | ±08.56 | ±06.77 |
| **RECTOR-L+** | ±08.82 | ±06.88 | **±04.86** | **±10.93** | **±08.29** | **±06.35** |

## C.5. AFP Initialization and Region-Granularity

We further analyze the sensitivity of AFP to the anatomical initialization and the number of learned regions $R$. Tables 8–9 show that RECTOR is not highly sensitive to moderate changes in $R$: performance remains competitive across a broad range of granularities, while the default setting is generally best or near-best. In contrast, replacing the anatomy-initialized partition with a random partition consistently degrades performance, indicating that AFP depends on meaningful partition structure rather than region count alone. Together, these results suggest that the anatomical prior mainly serves as a useful structural scaffold, while the final partition remains meaningfully adaptive rather than fixed by the initialization.

*Table 8.* **Sensitivity to the number of regions $R$ in AFP on SEED, SEED-IV, and DEAP.** (w-F1: Weighted-F1 (%); BAcc: Balanced Accuracy (%)). We vary the number of learned regions from {6, 9, 12}, and additionally compare against a 9-region random partition setting. (Random): Random region-channel initialization. SD: Subject Dependent; SI: Subject Independent. **BOLD**/UNDERLINE indicate the best/second-best results. * denotes RECTOR significantly outperforms its variant that uses random region-channel initialization.

| No. of Regions | SEED | | | | SEED-IV | | | | DEAP-Valence | | | | DEAP-Arousal | | | |
|---|---|---|---|---|---|---|---|---|---|---|---|---|---|---|---|---|
| | SD | | SI | | SD | | SI | | SD | | SI | | SD | | SI | |
| | w-F1 | BAcc | w-F1 | BAcc | w-F1 | BAcc | w-F1 | BAcc | w-F1 | BAcc | w-F1 | BAcc | w-F1 | BAcc | w-F1 | BAcc |
| 6 | $\mathbf{85.3}_{\pm08.4}$ | $\mathbf{84.7}_{\pm08.2}$ | $\mathbf{61.3}_{\pm08.6}$ | $\underline{61.0}_{\pm09.1}$ | $\underline{63.3}_{\pm11.3}$ | $62.2_{\pm11.8}$ | $\mathbf{44.9}_{\pm06.8}$ | $\mathbf{44.1}_{\pm07.1}$ | $\underline{69.0}_{\pm10.5}$ | $67.3_{\pm10.3}$ | $\underline{66.3}_{\pm11.3}$ | $\underline{65.5}_{\pm11.0}$ | $\mathbf{69.7}_{\pm12.2}$ | $\underline{67.5}_{\pm12.8}$ | $\underline{68.0}_{\pm12.0}$ | $\mathbf{67.7}_{\pm12.8}$ |
| 9 (Random) | $80.9_{\pm09.9}$ | $80.4_{\pm09.8}$ | $59.6_{\pm10.5}$ | $59.2_{\pm11.0}$ | $61.1_{\pm13.3}$ | $59.9_{\pm12.4}$ | $43.5_{\pm08.2}$ | $41.8_{\pm08.6}$ | $66.4_{\pm12.5}$ | $64.9_{\pm12.9}$ | $64.8_{\pm12.9}$ | $64.0_{\pm12.8}$ | $66.2_{\pm14.8}$ | $64.9_{\pm14.1}$ | $66.2_{\pm15.5}$ | $65.4_{\pm15.2}$ |
| 9 (RECTOR) | $\underline{85.0}^{*}_{\pm08.5}$ | $\mathbf{84.7}^{*}_{\pm08.7}$ | $\underline{61.1}^{*}_{\pm08.5}$ | $\mathbf{61.2}^{*}_{\pm08.8}$ | $\mathbf{63.7}^{*}_{\pm11.0}$ | $\underline{62.4}^{*}_{\pm11.4}$ | $\underline{44.8}^{*}_{\pm06.7}$ | $\underline{43.7}^{*}_{\pm06.8}$ | $\mathbf{69.4}^{*}_{\pm09.8}$ | $\mathbf{67.9}^{*}_{\pm10.1}$ | $\mathbf{66.7}^{*}_{\pm10.9}$ | $\mathbf{65.8}^{*}_{\pm11.2}$ | $\underline{69.7}^{*}_{\pm12.2}$ | $\mathbf{67.8}^{*}_{\pm12.4}$ | $\mathbf{68.4}^{*}_{\pm12.1}$ | $\underline{67.5}^{*}_{\pm13.3}$ |
| 12 | $84.5_{\pm08.6}$ | $\underline{84.4}_{\pm08.3}$ | $61.1_{\pm08.5}$ | $61.0_{\pm09.3}$ | $63.2_{\pm11.2}$ | $\mathbf{62.6}_{\pm11.7}$ | $44.5_{\pm06.9}$ | $43.0_{\pm07.1}$ | $68.4_{\pm10.2}$ | $\underline{67.4}_{\pm10.4}$ | $66.1_{\pm11.2}$ | $65.3_{\pm11.7}$ | $\underline{69.0}_{\pm12.4}$ | $67.4_{\pm12.5}$ | $67.6_{\pm12.1}$ | $67.3_{\pm13.6}$ |

*Table 9.* **Sensitivity to the number of regions $R$ in AFP on MSIT and ECR.** (AUROC (%); BAcc: Balanced Accuracy (%)). We vary the number of learned regions from {15, 30}, and additionally compare against a 30-region random partition setting. The 15-region partition is built by merging the left and right hemisphere regions from the 30-region setting. (Random): Random region-channel initialization. **BOLD**/UNDERLINE indicates the best/second-best results. * denotes RECTOR significantly outperforms its variant that uses random region-channel initialization.

| No. of Regions | MSIT | | ECR | |
|---|---|---|---|---|
| | AUROC | BAcc | AUROC | BAcc |
| 15 | $\underline{91.2}_{\pm04.6}$ | $\underline{83.6}_{\pm06.8}$ | $\underline{91.7}_{\pm04.6}$ | $\underline{83.6}_{\pm07.0}$ |
| 30 (Random) | $90.0_{\pm05.8}$ | $82.7_{\pm07.5}$ | $90.3_{\pm05.9}$ | $82.4_{\pm08.3}$ |
| 30 (RECTOR) | $\mathbf{92.1}^{*}_{\pm04.1}$ | $\mathbf{84.7}^{*}_{\pm06.4}$ | $\mathbf{92.4}^{*}_{\pm03.9}$ | $\mathbf{84.2}^{*}_{\pm06.6}$ |

## C.6. Stability and Non-Anatomical Refinement of AFP

Table 10 shows that AFP reassigns a substantial fraction of channels from the anatomical initialization, confirming that it is not trivially recovering the prior; on SEED and SEED-IV, these reassigned channels are also highly consistent across sessions. Table 11 and Fig. 10 further show that the learned partitions are highly stable across random seeds, slightly more variable across sessions of the same subject, and most variable across subjects, while still preserving a consistent macro-scale organization. These results support that AFP learns reproducible, spatially coherent functional refinements rather than arbitrary partitions.

*Table 10.* **Quantifying non-anatomical refinement learned by AFP.** We report the percentage of channels reassigned from the anatomical prior, and percentage of those reassigned channels that are consistent across sessions. "—" indicates that session annotations are unavailable for that dataset.

| | SEED | SEED-IV | DEAP | MSIT | ECR |
|---|---|---|---|---|---|
| Ch. reassigned from anatomy | $44.2\%_{\pm05.3\%}$ | $42.8\%_{\pm05.0\%}$ | $36.5\%_{\pm03.9\%}$ | $50.7\%_{\pm06.1\%}$ | $51.5\%_{\pm06.2\%}$ |
| Reassigned Ch. consistent across Sess. | $84.7\%_{\pm02.9\%}$ | $86.1\%_{\pm02.6\%}$ | — | — | — |

*Table 11.* **Stability of AFP partitions across random seeds, sessions, and subjects.** We report the percentage of channels whose assigned region differs from the corresponding consensus partition. Across all datasets, variability is lowest across random seeds, slightly higher across sessions of the same subject, and highest across subjects, indicating that AFP learns reproducible macro-scale topologies rather than arbitrary partitions. "—" indicates that session annotations are unavailable for that dataset.

| Comparison | SEED | SEED-IV | DEAP | MSIT | ECR |
|---|---|---|---|---|---|
| Across seeds | $02.3\%_{\pm00.9\%}$ | $02.9\%_{\pm01.2\%}$ | $03.1\%_{\pm01.0\%}$ | $03.9\%_{\pm01.4\%}$ | $04.0\%_{\pm01.6\%}$ |
| Across sessions | $05.1\%_{\pm01.8\%}$ | $05.5\%_{\pm01.7\%}$ | — | — | — |
| Across subjects | $10.1\%_{\pm02.7\%}$ | $11.2\%_{\pm03.3\%}$ | $12.3\%_{\pm03.1\%}$ | $13.6\%_{\pm03.5\%}$ | $14.2\%_{\pm03.6\%}$ |

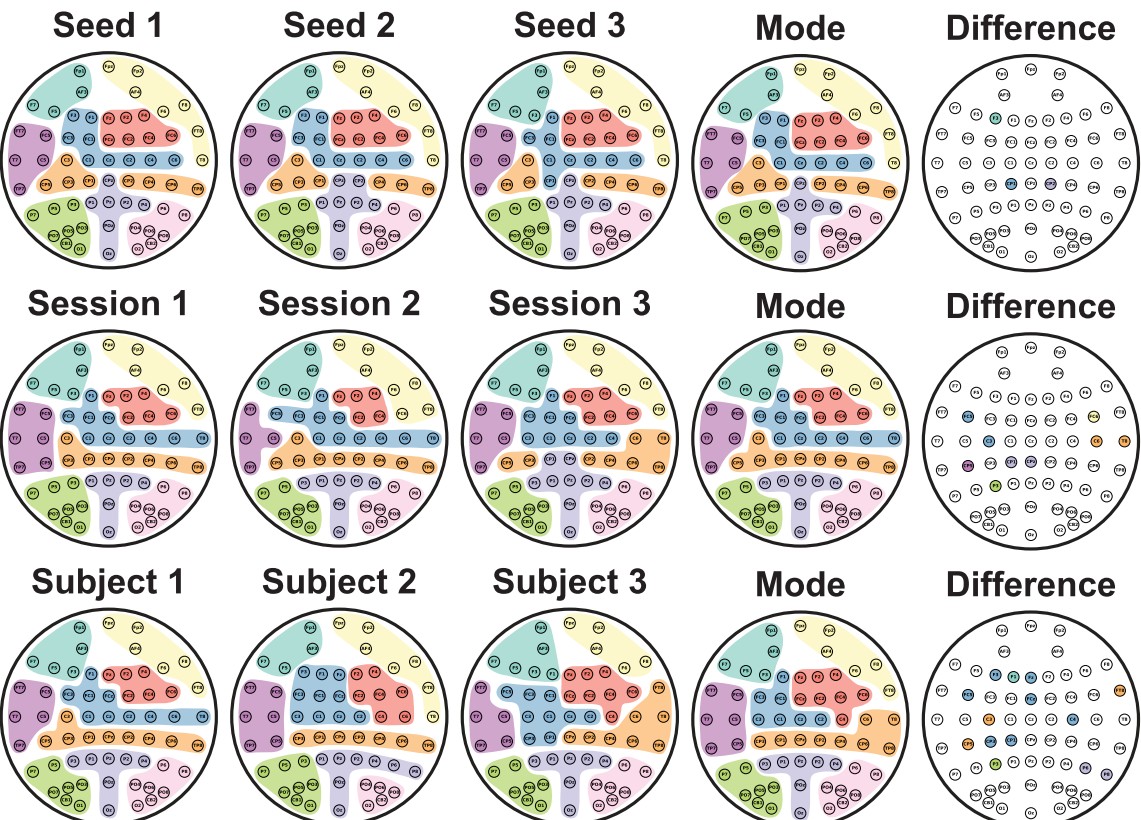

*Figure 10.* **Qualitative stability of AFP partitions across seeds, sessions, and subjects on SEED.** Each row shows consensus region-channel partitions for 3 random seeds (top), 3 sessions of the same subject (middle), and 3 representative subjects (bottom). The fourth column (Mode) shows the consensus partition across the corresponding seeds/sessions/subjects, and the last column highlights channels whose assignments differ from that consensus. Across seeds and sessions, the large-scale functional organization is highly consistent, while cross-subject differences remain modest and structured, indicating that AFP converges to reproducible macro-scale topologies.

### C.7. Sensitivity to the AFP Annealing Policy

Because AFP evolves from an anatomical prior to a learned functional topology through the annealing schedule $\alpha(t)$, we also evaluate how sensitive RECTOR is to this design choice. Tables 12–13 show that the model is not highly sensitive to nearby schedules, but gradual annealing is consistently preferable: the default cosine schedule achieves the best or near-best performance across EEG and sEEG benchmarks, whereas degenerate settings such as constant $\alpha(t) = 0$ are clearly worse. Fig. 11 provides a qualitative view of the same effect: keeping $\alpha(t)$ fixed overly constrains the learned partition by anatomy, while gradual annealing yields a smoother and more coherent functional refinement. Overall, these results indicate that RECTOR benefits from progressive relaxation of the anatomical prior, but does not require delicate schedule tuning in practice.

*Table 12.* **Sensitivity to the annealing policy $\alpha(t)$ in AFP on SEED, SEED-IV, and DEAP.** (w-F1: Weighted-F1 (%); BAcc: Balanced Accuracy (%)). We compare constant 0, constant 1, step 1→0 at epoch 100, linear decay, and the default cosine decay. SD: Subject Dependent; SI: Subject Independent. **BOLD**/UNDERLINE indicate the best/second-best results. * denotes RECTOR using cosine annealing significantly outperforms this annealing strategy.

| $\alpha(t)$ Annealing | SEED SD w-F1 | SEED SD BAcc | SEED SI w-F1 | SEED SI BAcc | SEED-IV SD w-F1 | SEED-IV SD BAcc | SEED-IV SI w-F1 | SEED-IV SI BAcc | DEAP-Valence SD w-F1 | DEAP-Valence SD BAcc | DEAP-Valence SI w-F1 | DEAP-Valence SI BAcc | DEAP-Arousal SD w-F1 | DEAP-Arousal SD BAcc | DEAP-Arousal SI w-F1 | DEAP-Arousal SI BAcc |
|---|---|---|---|---|---|---|---|---|---|---|---|---|---|---|---|---|
| Constant 0 | $81.4^*_{\pm09.4}$ | $80.8^*_{\pm09.6}$ | $59.3^*_{\pm09.8}$ | $58.5^*_{\pm09.2}$ | $61.6^*_{\pm12.0}$ | $60.3^*_{\pm12.7}$ | $44.0_{\pm07.6}$ | $42.2^*_{\pm07.3}$ | $66.9^*_{\pm11.1}$ | $65.4^*_{\pm11.2}$ | $65.2^*_{\pm12.4}$ | $64.1^*_{\pm12.3}$ | $66.0^*_{\pm13.4}$ | $64.9^*_{\pm13.0}$ | $66.8^*_{\pm13.9}$ | $65.9^*_{\pm14.1}$ |
| Constant 1 | $83.3^*_{\pm08.6}$ | $83.1^*_{\pm08.7}$ | $60.3^*_{\pm08.5}$ | $60.3^*_{\pm09.3}$ | $62.5^*_{\pm11.8}$ | $61.3^*_{\pm11.3}$ | $44.1_{\pm06.9}$ | $43.5^*_{\pm07.2}$ | $68.0^*_{\pm10.7}$ | $66.4^*_{\pm10.5}$ | $65.2^*_{\pm11.1}$ | $64.8^*_{\pm11.3}$ | $68.6^*_{\pm12.7}$ | $66.3^*_{\pm12.3}$ | $67.0^*_{\pm12.0}$ | $66.3^*_{\pm12.9}$ |
| 1→0 (100 Eps.) | $83.8^*_{\pm09.6}$ | $83.6^*_{\pm09.0}$ | $60.5^*_{\pm09.2}$ | $60.2_{\pm09.1}$ | $63.3_{\pm11.8}$ | $62.2_{\pm11.1}$ | $44.2_{\pm07.1}$ | $43.6_{\pm07.1}$ | $68.2^*_{\pm10.2}$ | $66.8^*_{\pm10.5}$ | $66.0_{\pm11.2}$ | $64.8_{\pm11.5}$ | $69.1_{\pm13.7}$ | $66.4^*_{\pm12.4}$ | $67.6_{\pm13.1}$ | $66.9_{\pm13.8}$ |
| Linear | $84.4_{\pm09.0}$ | $84.0_{\pm09.1}$ | **$61.2_{\pm09.1}$** | $61.0_{\pm08.7}$ | $63.1_{\pm11.5}$ | $62.1_{\pm11.7}$ | **$44.8^*_{\pm07.3}$** | **$43.9_{\pm06.8}$** | $68.8_{\pm09.4}$ | $67.2_{\pm11.1}$ | $66.3_{\pm10.3}$ | $65.2_{\pm11.7}$ | $69.2_{\pm13.0}$ | $67.0_{\pm12.2}$ | $68.1_{\pm12.2}$ | **$67.5_{\pm12.8}$** |
| Cosine (RECTOR) | **$85.0_{\pm08.5}$** | **$84.7_{\pm08.7}$** | $61.1_{\pm08.5}$ | **$61.2_{\pm08.8}$** | **$63.7_{\pm11.0}$** | **$62.4_{\pm11.4}$** | **$44.8_{\pm06.7}$** | $43.7_{\pm06.8}$ | **$69.4_{\pm09.8}$** | **$67.9_{\pm10.1}$** | **$66.7_{\pm10.9}$** | **$65.8_{\pm11.2}$** | **$69.7_{\pm12.2}$** | **$67.8_{\pm12.4}$** | **$68.4_{\pm12.1}$** | $67.5_{\pm13.3}$ |

*Table 13.* **Sensitivity to the annealing policy $\alpha(t)$ in AFP on MSIT and ECR.** (AUROC (%); BAcc: Balanced Accuracy (%)). We compare constant 0, constant 1, step 1→0 at epoch 100, linear decay, and the default cosine decay. **BOLD**/UNDERLINE indicate the best/second-best results. * denotes RECTOR using cosine annealing significantly outperforms this annealing strategy.

| $\alpha(t)$ Annealing | MSIT AUROC | MSIT BAcc | ECR AUROC | ECR BAcc |
|---|---|---|---|---|
| Constant 0 | $90.2^*_{\pm05.2}$ | $83.0^*_{\pm06.2}$ | $90.4^*_{\pm04.7}$ | $82.6^*_{\pm06.1}$ |
| Constant 1 | $91.1^*_{\pm04.9}$ | $83.6^*_{\pm06.0}$ | $91.2^*_{\pm04.7}$ | $83.5_{\pm06.3}$ |
| 1→0 (100 Eps.) | $91.5_{\pm04.5}$ | $83.8_{\pm06.8}$ | $91.3^*_{\pm04.2}$ | $83.0^*_{\pm06.4}$ |
| Linear | $91.9_{\pm05.1}$ | $84.7_{\pm06.3}$ | $91.8_{\pm04.0}$ | $83.8_{\pm06.9}$ |
| Cosine (RECTOR) | **$92.1_{\pm04.1}$** | **$84.7_{\pm06.4}$** | **$92.4_{\pm03.9}$** | **$84.2_{\pm06.6}$** |

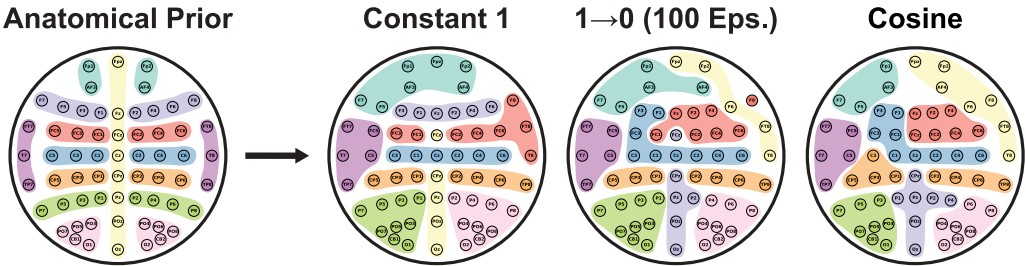

*Figure 11.* **Effect of the annealing policy $\alpha(t)$ on the learned AFP topology.** Starting from the same anatomical prior (left), we visualize the consensus region-channel partition learned under three annealing strategies on SEED. Constant $\alpha(t) = 1$ keeps the partition overly constrained by anatomy, a hard step $1 \rightarrow 0$ after 100 epochs permits adaptation but yields a slightly less smooth topology, and cosine annealing produces the most coherent functional refinement. This qualitatively matches the downstream ablation results and shows that gradual annealing leads to better topology formation.

### C.8. Optimization Stability

Fig. 12 shows the total SSL loss and the main unweighted loss components across five random seeds for both EEG (SEED) and sEEG (MSIT). In all cases, the total loss decreases smoothly, the individual loss terms remain well behaved, and the variation across seeds is small, with no evidence of divergence, collapse, or unstable topology learning. This supports that RECTOR's optimization remains stable in practice despite the adaptive discrete assignment mechanism.

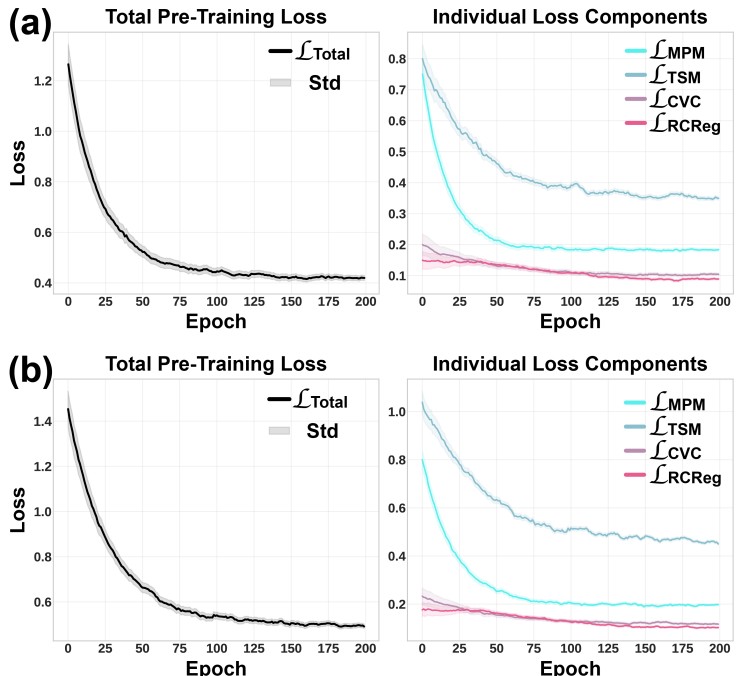

*Figure 12.* **Pre-training loss curves.** Total SSL loss (left) and the main unweighted loss components (right), shown as mean across 5 random seeds with shaded standard deviation. (a) EEG (SEED) and (b) sEEG (MSIT).

## D. Datasets

**MSIT & ECR**   The MSIT and ECR datasets (Provenza et al., 2019) probe human cognitive and emotional conflict responses using intracranial electrophysiological recordings from 17 participants with pharmaco-resistant partial seizures. Each subject performed two distinct conflict-based tasks: the Multi-Source Interference Task (MSIT), in which they identified a target number among distractors under varying congruency, and the Emotional Conflict Resolution (ECR) task, which required resolving conflict between facial expressions and superimposed emotional words (Fig. 13). Local field potentials (LFPs) were recorded via depth electrodes implanted across up to 30 anatomically defined brain regions, yielding 64–195 channels per participant. These LFP recordings were then used to classify rest-state versus task-state activity. By combining richly sampled, multi-site LFPs with behaviorally precise conflict paradigms, these datasets offer a unique window into both task-specific and generalizable neural mechanisms–insights that could guide the development of adaptive deep-brain stimulation therapies for cognitive disorders.

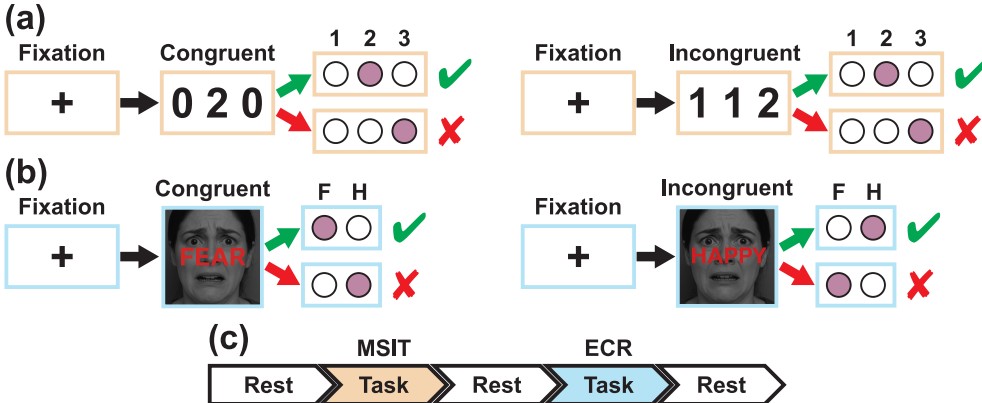

*Figure 13.* Behavioral paradigms for the MSIT and ECR tasks. (a) **MSIT**: Participants view three simultaneously presented digits and must press the button matching the value of the unique target digit, ignoring its position. Distractors are either zeros (congruent trial) or valid digits (incongruent trial). (b) **ECR**: Participants see an emotional word superimposed on a facial expression and must select the emotion denoted by the word, disregarding the face. Trials are congruent when word and expression match, and incongruent otherwise. (c) All trials are annotated as either rest or task (MSIT/ECR) for subsequent classification.

We adopt a subject-dependent evaluation paradigm. Each recording session is chronologically split into 80% training + validation sets (with 72% used for training and a random 8% for validation) and a 20% unshuffled hold-out test set. We insert a 4-second buffer between the training and test splits to ensure that no adjacent segments appear in both sets. Hyperparameters were chosen based on performance on this validation set. This approach prevents leakage of temporally adjacent sEEG patterns into both train and test sets and better simulates real-world deployment, where one must train on past data and predict on future recordings (future recordings could include information from the past data and cause implicit data leakage). LFPs were recorded at 2 kHz and downsampled to 1 kHz. We applied a 1–150 Hz band-pass filter to remove low- and high-frequency noise, and notch filters at 60 Hz and its harmonics to eliminate line noise. Before training, we segmented each trial into non-overlapping 4-second epochs and treated each epoch as an independent sample for classification.

**SEED & SEED-IV**  The SEED dataset (Zheng & Lu, 2015) comprises 62-channel EEG recordings from 15 participants who watched 15 Chinese film clips per session, each designed to elicit one of three emotional states (positive, neutral, negative). Each subject completed three sessions (45 trials in total). SEED-IV (Zheng et al., 2018) retains the same 62-channel, 15-subject, three-session design, but extends to four emotion classes (happy, neutral, sad, fear) with 24 trials per session (72 trials per subject).

We adopt two evaluation paradigms: (1) Subject-dependent (SD): experiments are conducted per session. For SEED, each session's 15 trials are split into the first 9 for training and the last 6 for testing, with 10% of the training samples as a validation set; for SEED-IV, each session's 24 trials are split into the first 16 for training and the remaining 8 for testing. (2) Subject independent (SI): for both SEED and SEED-IV, we perform leave-one-subject-out (LOSO) cross-validation across all subjects. For each fold of the LOSO, the remaining subjects form the full training set. This full training set is then randomly split into a 90% training partition and a 10% validation partition in a trial-wise manner. We train the model on the 90% partition and use the 10% validation partition for hyperparameter tuning. Therefore, the validation set is hold-out as it is unseen by the trained model. The final model is then evaluated on the entirely unseen test subject. All data were recorded at 1 kHz and downsampled to 200 Hz, then band-pass filtered between 1–50 Hz to remove artifacts. Prior to training, each trial is divided into non-overlapping 4-second segments, and each segment is treated as an independent sample for emotion classification. We avoided LDS smoothing due to its potential to introduce data leakage by incorporating future temporal information, which would compromise the integrity of the predictive task.

**DEAP**  The DEAP dataset (Koelstra et al., 2011) captures 32-channel EEG recordings from 32 participants as they watched 40 one-minute music video excerpts (40 trials). After each clip, participants provided subjective ratings on arousal, valence, dominance, liking, and familiarity, yielding rich physiological–affective annotations for downstream modeling.

We evaluate our models under two standard protocols: (1) Subject-dependent (SD): trial-wise 10-fold cross-validation for each subject, with 10% of the training samples as a validation set in each fold. (2) Subject independent (SI): leave-one-subject-out (LOSO) cross-validation across all subjects. All EEG recordings were acquired with a 32-electrode montage at 512 Hz and downsampled to 128 Hz. A 3-second pre-trial baseline was removed from each trial, and the data were band-pass filtered between 4-45 Hz to suppress low- and high-frequency noise. For affective labels (valence and arousal), participants rated each dimension on a 1–9 scale; we binarize these into "low" ($\leq 5$) and "high" ($> 5$) classes for downstream classification.

**Trial-Wise Splitting**  For our subject-dependent protocols, randomly shuffling the segments across trials before splitting into train and test sets can inadvertently place highly correlated adjacent segments into both splits, inflating measured accuracy. In real-world deployment, however, such overlap rarely occurs, which can lead to lower measured performance. To obtain a more realistic assessment, we instead split data at the trial level, ensuring that all segments from any given trial reside entirely in either the training or testing set, thereby preventing information leakage and yielding a more generalizable evaluation.

## E. Electrode and Region Maps of sEEG

For the sEEG data in MSIT and ECR datasets, each subject was implanted with bilateral depth electrodes, each comprising 8–16 contacts. Subjects had 5–9 electrodes in the right hemisphere, and 5–8 electrodes in the left hemisphere, yielding 64–195 bipolar-referenced channels per subject. Each channel is assigned to one of the 30 brain regions using the Electrode Labeling Algorithm (Peled et al., 2017). A detailed breakdown of channels per region is provided in Table 14.

*Table 14.* Number of sEEG channels per brain region across 17 subjects (Provenza et al., 2019).

| Region | Subject | | | | | | | | | | | | | | | | |
|---|---|---|---|---|---|---|---|---|---|---|---|---|---|---|---|---|---|
| | 1 | 2 | 3 | 4 | 5 | 6 | 7 | 8 | 9 | 10 | 11 | 12 | 13 | 14 | 15 | 16 | 17 |
| L-NAcc | 0 | 0 | 0 | 0 | 2 | 0 | 0 | 0 | 1 | 0 | 1 | 0 | 1 | 0 | 2 | 0 | 1 |
| L-amyg | 4 | 4 | 4 | 5 | 2 | 5 | 3 | 5 | 0 | 2 | 4 | 5 | 4 | 0 | 1 | 6 | 10 |
| L-caudate | 0 | 0 | 0 | 0 | 7 | 3 | 10 | 2 | 7 | 0 | 7 | 5 | 2 | 3 | 9 | 8 | 7 |
| L-hipp | 5 | 3 | 2 | 7 | 4 | 10 | 4 | 0 | 0 | 6 | 11 | 10 | 5 | 6 | 6 | 8 | 2 |
| L-dACC | 1 | 1 | 1 | 3 | 0 | 5 | 0 | 0 | 4 | 2 | 6 | 4 | 4 | 4 | 5 | 8 | 0 |
| L-dlPFC | 8 | 11 | 10 | 15 | 15 | 31 | 17 | 16 | 43 | 33 | 35 | 19 | 18 | 30 | 29 | 23 | 12 |
| L-dmPFC | 7 | 3 | 1 | 3 | 3 | 2 | 10 | 19 | 1 | 4 | 6 | 5 | 1 | 3 | 4 | 9 | 5 |
| L-insula | 0 | 0 | 0 | 1 | 0 | 0 | 0 | 2 | 3 | 0 | 0 | 0 | 0 | 0 | 0 | 0 | 0 |
| L-lOFC | 4 | 4 | 0 | 7 | 8 | 7 | 10 | 8 | 8 | 4 | 8 | 9 | 9 | 8 | 9 | 7 | 7 |
| L-mOFC | 2 | 1 | 2 | 2 | 1 | 3 | 4 | 1 | 1 | 3 | 1 | 0 | 0 | 1 | 0 | 1 | 0 |
| L-parahipp | 0 | 0 | 0 | 0 | 0 | 0 | 0 | 3 | 0 | 0 | 0 | 0 | 2 | 0 | 0 | 0 | 0 |
| L-postCC | 0 | 2 | 1 | 0 | 0 | 0 | 0 | 0 | 0 | 0 | 0 | 1 | 1 | 0 | 0 | 0 | 0 |
| L-rACC | 0 | 1 | 0 | 0 | 3 | 0 | 0 | 0 | 0 | 0 | 3 | 5 | 0 | 4 | 1 | 1 | 0 |
| L-temporal | 6 | 5 | 6 | 10 | 19 | 22 | 17 | 13 | 13 | 19 | 22 | 26 | 24 | 26 | 31 | 23 | 11 |
| L-vlPFC | 1 | 0 | 0 | 9 | 12 | 4 | 6 | 11 | 1 | 4 | 4 | 6 | 4 | 6 | 6 | 6 | 6 |
| R-NAcc | 0 | 0 | 0 | 0 | 3 | 0 | 0 | 0 | 0 | 0 | 2 | 0 | 1 | 0 | 0 | 0 | 0 |
| R-amyg | 4 | 3 | 1 | 5 | 5 | 0 | 0 | 0 | 2 | 5 | 5 | 0 | 2 | 6 | 0 | 3 | 5 |
| R-caudate | 0 | 0 | 1 | 8 | 6 | 0 | 0 | 0 | 0 | 0 | 6 | 1 | 1 | 9 | 7 | 0 | 7 |
| R-hipp | 2 | 2 | 5 | 3 | 7 | 3 | 0 | 1 | 6 | 11 | 7 | 15 | 7 | 13 | 5 | 6 | 7 |
| R-dACC | 0 | 0 | 2 | 1 | 1 | 6 | 1 | 0 | 2 | 2 | 7 | 4 | 6 | 1 | 2 | 3 | 0 |
| R-dlPFC | 6 | 9 | 14 | 23 | 12 | 7 | 2 | 1 | 10 | 24 | 20 | 24 | 8 | 29 | 29 | 18 | 6 |
| R-dmPFC | 7 | 5 | 3 | 12 | 3 | 1 | 10 | 0 | 1 | 8 | 0 | 2 | 0 | 9 | 6 | 5 | 10 |
| R-insula | 0 | 0 | 0 | 0 | 0 | 0 | 0 | 2 | 0 | 0 | 0 | 0 | 0 | 0 | 0 | 0 | 0 |
| R-lOFC | 5 | 6 | 3 | 8 | 8 | 2 | 15 | 13 | 3 | 10 | 7 | 8 | 7 | 3 | 8 | 6 | 6 |
| R-mOFC | 0 | 1 | 0 | 3 | 1 | 0 | 4 | 2 | 1 | 2 | 2 | 0 | 4 | 0 | 1 | 1 | 1 |
| R-parahipp | 0 | 0 | 0 | 0 | 0 | 0 | 0 | 3 | 0 | 0 | 0 | 0 | 5 | 0 | 0 | 0 | 1 |
| R-postCC | 0 | 3 | 0 | 0 | 0 | 0 | 0 | 0 | 0 | 0 | 0 | 0 | 0 | 3 | 1 | 0 | 0 |
| R-rACC | 0 | 0 | 0 | 1 | 0 | 0 | 1 | 1 | 1 | 0 | 3 | 2 | 0 | 0 | 4 | 0 | 1 |
| R-temporal | 4 | 7 | 5 | 12 | 16 | 16 | 0 | 21 | 10 | 18 | 8 | 24 | 11 | 22 | 23 | 13 | 19 |
| R-vlPFC | 3 | 0 | 3 | 12 | 3 | 12 | 8 | 15 | 2 | 5 | 14 | 8 | 4 | 8 | 6 | 8 | 6 |
| Total | 69 | 71 | 64 | 150 | 141 | 139 | 122 | 139 | 120 | 162 | 189 | 183 | 131 | 194 | 195 | 163 | 130 |

# F. Implementation Details

## F.1. Model Configurations

RECTOR-Transformer block is used in the encoder, predictor, and decoder. The detailed configurations can be found in Table 15.

*Table 15.* Model configurations. RECTOR-L in ().

| Parameter | Encoder | Predictor | Decoder |
|---|---|---|---|
| RECTOR-Transformer block | 8 | 2 | 4 |
| Hidden dimension | 64 (128) | 64 (128) | 64 (128) |
| Head | 4 | 4 | 4 |
| Feed-forward dimension | 256 (512) | 256 (512) | 256 (512) |

## F.2. Positional Embeddings

To better capture both temporal dependency and spatial topology, we incorporate Rotary Position Embeddings (RoPE) on the temporal dimension (Su et al., 2024), and learnable positional embeddings on the spatial axis (Dwivedi & Bresson,

2020).

## F.3. Interpretability Maps

We computed Grad-CAM (Selvaraju et al., 2017) saliency heatmaps for each emotion class. Specifically, after passing an input trial through the model, we applied Grad-CAM to the output layer (before the classification head) to produce a per-channel/region importance score indicating how strongly each input location contributed to the predicted class. We also extracted self-attention scores from RECTOR-SA. For each attention head, we computed the mean attention score between every pair of region/channel tokens across all time segments. All maps and attention summaries were averaged across subjects within each dataset.

## F.4. Pre-training on Foundation Models

For neural foundation models in this work, we used publicly available checkpoints and then pre-trained the model on the target dataset with the same number of iterations as RECTOR.

## F.5. Hyperparameter Settings

*Table 16.* Pre-training settings.

| Hyperparameter | Setting |
|---|---|
| Epochs | 200 |
| Warmup epochs | 40 |
| Batch size | 256 |
| Dropout | 0.3 |
| Optimizer | AdamW |
| $\beta$ | (0.9, 0.999) |
| Scheduler | Cosine Annealing Scheduler |
| Learning rate | 5e-4 |
| Minimal learning rate | 1e-6 |
| Weight decay | 5e-2 |
| EMA momentum schedule | Linear scheduler |
| EMA start momentum | 0.9 |
| EMA final momentum | 1.0 |
| Target mask ratio | 0.2/0.2/0.2/0.1/0.1 |
| Patch size | 0.5 second |
| $\lambda_1, \lambda_2, \lambda_3$ | 0.5, 0.5, 0.1 |
| $\alpha(t)$ | Cosine Annealing Scheduler from 1 to 0 |
| $\tau$ | 0.5 for 100 epochs, 1 for 100 epochs |

*Table 17.* Fine-tuning settings.

| Hyperparameter | Setting |
|---|---|
| Epochs | 50 |
| Warmup epochs | 10 |
| Batch size | 256 |
| Dropout | 0.3 |
| Optimizer | AdamW |
| $\beta$ | (0.9, 0.999) |
| Scheduler | Cosine Annealing Scheduler |
| Learning rate | 5e-4 |
| Minimal learning rate | 1e-6 |
| Weight decay | 5e-2 |
| Label smoothing | 0.1 |
| $\alpha(t)$ | 0 |
| $\tau$ | 1 |

### F.6. Pre-training Time

The pre-training time for RECTOR, RECTOR+, and RECTOR-L+ on various datasets is reported in Table 18.

*Table 18.* RECTOR's pre-training time on various datasets.

| Model | Dataset | Pre-Training Time |
|---|---|---|
| **RECTOR** | SEED | 58 mins |
| | SEED-IV | 56 mins |
| | DEAP | 28 mins |
| | MSIT | 53 mins |
| | ECR | 54 mins |
| **RECTOR+** | SEED+SEED-IV+DEAP+CHB-MIT | 10.6 hours |
| | MSIT+ECR | 2.0 hours |
| **RECTOR-L+** | SEED+SEED-IV+DEAP+CHB-MIT | 12 hours |
| | MSIT+ECR | 2.0 hours |

### F.7. Model Size

We provide a detailed model size comparison for all baselines and RECTOR variants for EEG and sEEG applications in Table 19. Our RECTOR and RECTOR-L are in a medium range in terms of model size. This analysis confirms that RECTOR achieves state-of-the-art performance primarily through architectural innovation, rather than scaling of parameters alone.

*Table 19.* Model size for baselines and RECTOR for EEG and sEEG datasets.

| Model | EEG Model Size | sEEG Model Size |
|---|---|---|
| TSception | 25K | 923K |
| Conformer | 281K | 4.6M |
| LGGNet | 166K | 15.2M |
| PGCN | 12.3M | |
| EmT | 704K | |
| MMM | 40K | |
| CBraMod | 4.0M | |
| PopT | 20M | 20M |
| Seegnificant | | 789K |
| Brant | | 500M |
| Du-IN | | 4.4M |
| BaRISTA | | 1M |
| LUNA | 7M | |
| mdJPT | 1M | |
| MAE | 311K | 345K |
| JEPA | 311K | 345K |
| CAE | 311K | 345K |
| **RECTOR** | 496K | 570K |
| **RECTOR-L** | 1.9M | 2.1M |

### F.8. Channel Matching for Expanded Pre-Training

To accommodate differing channel montages during expanded pre-training, we employ two strategies:

1. SEED/SEED-IV → DEAP (62 → 32 channels): We restrict the encoder to use the 32 channels common to both datasets, preserving a consistent 32-channel architecture throughout pre-training and fine-tuning.

2. DEAP → SEED/SEED-IV (32 → 62 channels): We use a 62-channel encoder augmented with 30 null tokens. An attention mask prevents any interaction with these null tokens during pre-training on DEAP. When pre-training returns to SEED/SEED-IV, we remove the mask and activate all 62 channels for full token mixing.

### F.9. Compute Resources

All experiments were conducted using Python 3.8.16 and PyTorch 2.0.1 with CUDA 11.7 for GPU acceleration. The primary training environment ran on Red Hat Enterprise Linux 7.9 with AMD Ryzen Threadripper PRO 5995WX 64-Cores CPU, and $2 \times$ NVIDIA GeForce RTX 4090 24 GB.

### F.10. Pseudocode

We decompose RECTOR into two main stages and present pseudocode for each: self-supervised pre-training in Algorithm 1 and downstream fine-tuning in Algorithm 2. Together, these algorithms illustrate the end-to-end training pipeline of RECTOR.

---

**Algorithm 1** Self-Supervised Pre-training of RECTOR

---

**Input:** batch $\mathbf{X}^{\mathrm{C}}$, context encoder $f(\cdot; \theta)$, target encoder $\tilde{f}(\cdot; \tilde{\theta})$, predictor $g(\cdot; \psi)$, decoder $h(\cdot; \phi)$, learning rate $\alpha$, EMA rate $\beta$, projection matrices
**Output:** Updated encoder $f(\cdot; \theta)$
**for** each $\mathbf{X}^{\mathrm{C}}$ **do**
    *// (a) Construct assignment matrix*
    $\mathbf{\Pi}^{\mathbf{X}} \leftarrow \Lambda(\mathbf{X}^{\mathrm{C}}, \mathbf{W}_{\mathbf{\Pi}}^{\mathbf{X}})$
    $\mathbf{S}^{\mathbf{X}} \leftarrow \mathrm{OneHot}(\mathrm{argmax}_r(\mathbf{\Pi}^{\mathbf{X}})_{:,r})$
    $\mathbf{S}_{\mathrm{STE}}^{\mathbf{X}} \leftarrow \mathbf{S}^{\mathbf{X}} + \mathrm{sg}(\mathbf{\Pi}^{\mathbf{X}} - \mathbf{S}^{\mathbf{X}})$
    $\mathbf{E}^{\mathrm{R}} \leftarrow \texttt{RegionalEmbedding}(\dots)$
    *// (b) Construct token sequence*
    **for** $t = 1$ **to** $T$ **do**
        $[\mathbf{X}^{\mathrm{R}}]_{t,:} \leftarrow \mathbf{\Pi}^{\mathbf{X}^{\top}}[\mathbf{X}^{\mathrm{C}}]_{t,:} + \mathbf{E}^{\mathrm{R}}$
    **end for**
    $\mathbf{X} \leftarrow [\mathbf{X}^{\mathrm{C}}; \mathbf{X}^{\mathrm{R}}]$
    *// (c) Sample context & target views*
    $\mathbf{M}_c, \{\mathbf{M}_m\} \leftarrow \texttt{StructuredMultiViewMasking}(\mathbf{X}, \dots)$
    $\mathbf{X}_c \leftarrow [\mathbf{M}_c \mathbf{X}^{\mathrm{C}}; \mathbf{X}^{\mathrm{R}}]$
    **for all** $m \in \mathcal{M}$ **do**
        $\mathbf{X}_m \leftarrow [\mathbf{M}_m \mathbf{X}^{\mathrm{C}}; \mathbf{X}^{\mathrm{R}}]$
    **end for**
    *// (d) Embeddings*
    $\mathbf{P}_{\mathbf{c}}, \{\mathbf{P}_m\}, \mathbf{P} \leftarrow \texttt{PositionalEmbedding}(\dots)$
    *// (e) Encode context & targets*
    $\mathbf{Z_c} \leftarrow f(\mathbf{X_c}, \mathbf{P_c}; \theta)$
    $\mathbf{Z} \leftarrow \tilde{f}(\mathbf{X}, \mathbf{P}; \tilde{\theta})$
    **for all** $m \in \mathcal{M}$ **do**
        $\mathbf{Z}_m \leftarrow \mathbf{M}_m \mathbf{Z}$
    **end for**
    *// (f) Predict & reconstruct*
    **for all** $m \in \mathcal{M}$ **do**
        $\hat{\mathbf{Z}}_m \leftarrow g(\mathbf{Z_c}, \mathbf{P}_m; \psi)$
        $\hat{\mathbf{X}}_m \leftarrow h(\hat{\mathbf{Z}}_m, \mathbf{P}_m; \phi)$
        $\hat{\mathbf{\Pi}}_m^{\mathbf{Z}} \leftarrow \Lambda(\hat{\mathbf{Z}}_m, \mathbf{W}_{\mathbf{\Pi}}^{\mathbf{Z}})$
        $\tilde{\mathbf{\Pi}}_m^{\mathbf{Z}} \leftarrow \mathbf{M}_m \Lambda(\mathbf{Z}, \mathbf{W}_{\mathbf{\Pi}}^{\mathbf{Z}})$
        $\mathbf{\Pi}_m^{\mathbf{X}} \leftarrow \Lambda(\mathbf{X}_m, \mathbf{W}_{\mathbf{\Pi}}^{\mathbf{X}})$
    **end for**
    *// (g) Compute losses*
    $\mathcal{L} \leftarrow 0$
    **for all** $m \in \mathcal{M}$ **do**
        $\mathcal{L} \leftarrow \mathcal{L} + \mathrm{MSE}(\hat{\mathbf{Z}}_m, \mathbf{Z}_m) + \mathrm{MSE}(\hat{\mathbf{X}}_m, \mathbf{X}_m) + \mathrm{CE}(\hat{\mathbf{\Pi}}_m^{\mathbf{Z}}, \tilde{\mathbf{\Pi}}_m^{\mathbf{Z}}) + \mathrm{CE}(\hat{\mathbf{\Pi}}_m^{\mathbf{Z}}, \mathbf{\Pi}_m^{\mathbf{X}})$
        **for all** $m' \neq m$ **do**
            $\check{\mathbf{z}}_m \leftarrow \mathrm{TokenMean}(\hat{\mathbf{Z}}_m)$
            $\check{\mathbf{z}}_{m'} \leftarrow \mathrm{TokenMean}(\hat{\mathbf{Z}}_{m'})$
            $\mathcal{L} \leftarrow \mathcal{L} + \mathrm{MSE}(\check{\mathbf{z}}_m, \check{\mathbf{z}}_{m'})$
        **end for**
    **end for**
    *// (h) Backprop & update*
    $\theta \leftarrow \theta - \alpha \nabla_\theta \mathcal{L}, \quad \psi \leftarrow \psi - \alpha \nabla_\psi \mathcal{L}, \quad \phi \leftarrow \phi - \alpha \nabla_\phi \mathcal{L}$
    $\tilde{\theta} \leftarrow \beta \tilde{\theta} + (1 - \beta)\theta$
**end for**

---

---

**Algorithm 2** Fine-Tuning RECTOR

---

**Input:** batch $(\mathbf{X}^{\mathrm{C}}, y)$, pre-trained encoder $f(\cdot)$, prediction head $\mathbf{W}_{\mathrm{out}}$, learning rate $\alpha$
**Output:** Updated encoder $f(\cdot)$, prediction head $\mathbf{W}_{\mathrm{out}}$
**for** each $(\mathbf{X}^{\mathrm{C}}, y)$ **do**
 *// (a) Construct assignment matrix*
 $\mathbf{\Pi}^{\mathbf{X}} \leftarrow \Lambda(\mathbf{X}^{\mathrm{C}}, \mathbf{W}_{\mathbf{\Pi}}^{\mathbf{X}})$
 $\mathbf{S}^{\mathbf{X}} \leftarrow \mathrm{OneHot}(\mathrm{argmax}_r(\mathbf{\Pi}^{\mathbf{X}})_{:,r})$
 $\mathbf{S}_{\mathrm{STE}}^{\mathbf{X}} \leftarrow \mathbf{S}^{\mathbf{X}} + \mathrm{sg}(\mathbf{\Pi}^{\mathbf{X}} - \mathbf{S}^{\mathbf{X}})$
 $\mathbf{E}^{\mathrm{R}} \leftarrow \mathtt{RegionalEmbedding}(\ldots)$
 $\mathbf{P} \leftarrow \mathtt{PositionalEmbedding}(\ldots)$
 *// (b) Construct token sequence*
 **for** $t = 1$ **to** $T$ **do**
  $[\mathbf{X}^{\mathrm{R}}]_{t,:} \leftarrow \mathbf{\Pi}^{\mathbf{X}^{\top}}[\mathbf{X}^{\mathrm{C}}]_{t,:} + \mathbf{E}^{\mathrm{R}}$
 **end for**
 $\mathbf{X} \leftarrow [\mathbf{X}^{\mathrm{C}}; \mathbf{X}^{\mathrm{R}}]$
 *// (c) Extract features*
 $\mathbf{Z} \leftarrow f(\mathbf{X}, \mathbf{P})$
 $h \leftarrow \mathrm{Flatten}(\mathbf{Z})\mathbf{W}_{\mathrm{out}}$
 *// (d) Compute loss, backprop & update*
 $\mathcal{L} \leftarrow \mathtt{CrossEntropy}(h, y)$
 Update all weights via $-\alpha \nabla \mathcal{L}$
**end for**

---

## G. Evaluation Metrics

We evaluated the performance of RECTOR and all baselines using the Area Under the Receiver Operating Characteristic Curve (AUROC) for task engagement classification on MSIT and ECR, and the Weighted F1 and Cohen's kappa (in Appendix) for emotion recognition on SEED, SEED-IV, and DEAP. Balanced accuracy is used for both tasks. In the subject-dependent setting, results are averaged per subject in MSIT, ECR, and DEAP, and across subjects and sessions in SEED and SEED-IV. In the subject-independent setting, results are averaged across subjects for every dataset. All metrics are reported as Mean$_{\pm\mathrm{Std}}$.

We indicated the statistical significance for classification performance using a Wilcoxon signed-rank test ($p < 0.05$). The test was applied across subjects for MSIT, ECR, and DEAP, and across both subjects and sessions for SEED and SEED-IV.

## H. Limitations

We do not interpret AFP-learned regions as definitive neuroscientific parcellations. Instead, they are best viewed as data-driven, task-adaptive grouping variables for structured attention that provide supportive evidence of neurophysiologically plausible organization within the learned representation. Accordingly, the interpretability results in Section 3.5 are not external clinical or neuroscientific validation, and we do not claim improvements in clinical trust or outcome from them alone. Finally, AFP currently relies on a hard one-hot channel-to-region assignment, which cannot represent overlapping functional memberships; extending AFP to mixed or soft overlapping assignments is an important future direction.

