# OpenReview forum: "RECTOR: Masked Region-Channel-Temporal Modeling for Affective and Cognitive Representation Learning"
_ICML.cc/2026/Conference — ICML 2026 regular_

### Official Review · Reviewer_qUWE · 2026-02-28

**Soundness:** 4
**Presentation:** 3
**Significance:** 4
**Originality:** 3
**Overall Recommendation:** 4
**Confidence:** 5

**Summary:**

the paper present RECTOR which is an end-to-end self-supervised framework that unifies joint region-channel-temporal representation learning beyond fixed anatomical priors to solve mssing channels and cross-montage generalization.

**Compliance With Llm Reviewing Policy:**

Affirmed.

**Final Justification:**

This is a solid paper, and the rebuttal has further strengthened the manuscript. However, when compared to similar works in the field, it does not demonstrate a significant breakthrough. Therefore, I do not feel a higher score is warranted at this time.

**Key Questions For Authors:**

1. How were the specific values for the various loss coefficients (e.g., lamda) determined?
2. To what extent does the initial anatomical prior influence the final functional partition and the model's convergence stability?
3. How to draw conclusions about sharper decay from Figure 7d?
4. Could you provide loss logs during pretraining?

**Limitations:**

Supplementing quantitative validation of interpretability related to neuroscience/clinical evidence.

**Strengths And Weaknesses:**

**Strength**
1. RECTOR sets a new benchmark in EEG emotion recognition and sEEG task-engagement classification, outperforming leading foundation models.
2. the Adaptive Functional Partitioning allows the model to evolve static definitions into task-specific functional regions and shows neurophysiologically visualizations.
3. Solid architecture ablation.
4. Performance consistently improves as the model is pre-trained on larger heterogeneous datasets, showing a clear scaling trend.

**Weakness**
1. The framework relies on a highly complex unified loss function combining four distinct components, which requires careful hyperparameter tuning for weights.
2. While the partitioning is adaptive, it still requires an initial anatomical prior to start the training process and no ablations about it.
3. The interpretability results (Figure 7) are not externally validated and most interpretability results are qualitative, not providing quantifications of improvements in clinical trust or outcome.
4. Some hyperparameters lack justification or analysis of sensitivity.

---

> ### Author Rebuttal · Authors · 2026-03-30
>
> We thank the reviewer for the thoughtful feedback and refer you to rebuttal supplement (Tables S1–S17, Figs. S1–S7): https://anonymous.4open.science/r/ICML2026_Rebuttal-6FDE/icml2026_rebuttal.pdf
> ## Re: W1, Q1
> We selected the loss coefficients through a sweep minimizing validation loss. The unified objective is controlled by only three coefficients; in the final model, we use $\lambda_1,\lambda_2,\lambda_3=0.5,0.5,0.1$ (Table 10). RCReg prevents trivial region/channel representations and topological collapse, so it acts as a lightweight regularizer
>
> Empirically, Table 3 shows these terms are not redundant: removing TSM, CVC, or either part of MPM consistently degrades downstream performance, with the largest drop when the broader MTRL objective is removed. Thus, the unified loss is a compact set of complementary objectives tuned by validation and supported by ablation
> ## Re: W2, Q2
> We added ablations in Fig. S1 and Tables S1–S2 to test how the anatomical prior constrains the final solution. Replacing anatomical initialization with a random one consistently hurts performance (e.g., SEED, 9-region random init: 80.9 vs 85.0 for anatomical init., w-F1, SD). Tables S1–S2 also show RECTOR is not overly sensitive to the anatomical prior: across anatomical granularities, performance remains competitive, with the default typically best/near-best and only very coarse/fine partitions slightly worse. This suggests that the anatomical prior mainly provides a useful structural scaffold
>
> Stds are also consistently larger under random initialization, suggesting weaker convergence stability (e.g., ECR, 30-region random vs. anatomical init.: AUROC ±5.9 vs ±3.9). Fig. S1 further shows that AFP reshapes the initial boundaries into smoother, data-driven groupings while preserving coherent large-scale organization. Thus, the anatomical prior affects where training starts and how stably it converges, but the final partition remains meaningfully adaptive rather than fixed by the prior
> ## Re: Q3
> We agree that “sharper decay” in Fig. 7(d) should be defined more explicitly. The Regional Importance Score (RIS) is normalized across regions and sum to 1. If $g^{r,k}$ is the region-level Grad-CAM [1] score of region $r$ for class $k$, we define $\text{RIS}^{r,k}=\frac{|g^{r,k}|}{\sum_{r’}|g^{r',k}|}$. RIS thus measures the relative share of class-discriminative evidence carried by each region
>
> Under this definition, sharper decay means the normalized importance mass is concentrated in fewer top-ranked regions. Thus, when RECTOR-AFP shows sharper decay than RECTOR-Fixed, the topology yields a more concentrated and discriminative attribution profile, consistent with “isolates discriminative functional regions”. We will clarify this in revision
> ## Re: W3, Limitations
> We agree that the interpretability results in Fig. 7 are not external clinical validation, and we do not claim improvements in clinical trust or outcome. Fig. 7 is intended as supportive evidence that the topology is structured and neurophysiologically plausible, and we will revise the paper to make this scope explicit
>
> Fig. 7 combines channel-level Grad-CAMs, region-channel attention structure, and RIS, and Appendix F.3 defines the Grad-CAM procedure. We also added quantitative evidence that AFP partitions are stable across random seeds, sessions, and subjects (Table S5/Fig. S4), remain largely preserved from pretraining to finetuning (Table S3/Fig. S2), and exhibit nontrivial but consistent refinement beyond fixed anatomy (Fig. S6/Table S16). These results strengthen the claim that the structures are robust and not merely qualitative visualizations
>
> Clinical outcome claims require a different level of validation and are beyond the scope of this ML paper. Our more modest claim is that interpretability may aid neuroscientific and clinical research by helping identify candidate biomarkers, compare learned groupings with anatomical circuits, and generate hypotheses about regions relevant for diagnosis or adaptive neurostimulation. We will revise the paper to present these as future directions, and add these discussions in the Limitations section
> ## Re: Q4
> We added pretraining loss logs in Fig. S7, showing total SSL loss and loss components, averaged over 5 seeds for SEED and MSIT. For further discussion, please see our response to Reviewer Ze3F, W3
> ## Re: W4
> We added sensitivity analyses for the key hyperparameters: Tables S1–S2 and Fig. S1 analyze the number of regions $R$, Tables S6–S7 and Fig. S5 analyze the annealing schedule, and Tables S8–S9 analyze the temperature. Across these studies, the model remains reasonably stable, with the default settings achieving the best/near-best performance. For further interpretation, see our responses to Reviewer YWsS, W1, Q1 ($R$) and Reviewer Ze3F, W4, Q1 ($\alpha(t)$, $\tau$). Other hyperparameters were chosen by validation-based tuning
> ## Reference
> [1] Grad-CAM: Visual Explanations from Deep Networks via Gradient-based Localization

---

> > ### Author Rebuttal · Reviewer_qUWE · 2026-04-02
> >
> > The authors solved my questions, and I maintain my positive recommendation.

---

> > > ### Author Response · Authors · 2026-04-02
> > >
> > > Thank you for your time and insightful feedback. We are thrilled that our response fully resolved your concerns, and we deeply appreciate your continued positive recommendation.
> > >
> > > Your constructive suggestions directly guided several key improvements that are now included in the rebuttal supplement and planned for the final manuscript:
> > >
> > > 1. **Model design and tuning**: Validation-based justification of the loss coefficients, plus sensitivity analyses for anatomical prior/random initialization, region granularity, annealing schedule, and temperature.
> > > 2. **Interpretability and topology validation**: An explicit definition of the Regional Importance Score, together with quantitative and qualitative evidence that AFP is stable across seeds, sessions, and subjects, remains largely preserved from pre-training to fine-tuning, and learns nontrivial refinements beyond fixed anatomy.
> > > 3. **Optimization stability**: Pre-training loss curves over 5 random seeds, showing smooth total-loss and component-loss trajectories.
> > >
> > >
> > > We believe your review has significantly strengthened the technical rigor of this work, and we hope these clarifications and new empirical results are helpful for your final evaluation.

---

### Official Review · Reviewer_Ze3F · 2026-03-12

**Soundness:** 3
**Presentation:** 3
**Significance:** 2
**Originality:** 3
**Overall Recommendation:** 4
**Confidence:** 3

**Summary:**

This paper introduces a self-supervised learning framework called RECTOR, designed to learn emotional and cognitive representations from electroencephalogram (EEG) and surface electroencephalogram (sEEG) signals. Its core architecture is RECTOR-SA, a block-sparse self-attention mechanism driving adaptive feature partitioning (AFP). AFP dynamically groups channels to functional regions, rather than strictly relying on fixed anatomical structures. During the pre-training phase, authors propose mask prediction modeling, topology modeling, and mask topology and representation learning (MTRL) to optimize cross-view consistency. The proposed method is evaluated on several downstream tasks, including EEG emotion recognition and sEEG task engagement classification, with good experimental results and robustness to channel loss.

**Compliance With Llm Reviewing Policy:**

Affirmed.

**Final Justification:**

My concerns have been resolved, according to the author's reply.

**Key Questions For Authors:**

- 1. How sensitive is the model's final performance to the specific decay strategy of $\alpha(t)$ in Equation 4? If $\alpha$ immediately drops to 0 or remains unchanged, will the learned topology deteriorate significantly?

 - 2. In a typical evaluation run, what is the empirical sparsity (the proportion of pruned connections) achieved by the local and global feature attention modules through top-p gating?

 - 3. Given the significant differences in the number of channels per region in sEEG data, how does this imbalance affect the actual hardware utilization and computational efficiency of the block sparse local feature attention?

**Limitations:**

See Weaknesses.

**Strengths And Weaknesses:**

**Strengths:**
The framework proposed in this paper facilitates the transition from static anatomical priors to adaptive, data-driven functional regions through the AFP module, providing a structurally sound and fundamental approach for dynamic brain network modeling. Its comprehensive experimental evaluation covers both non-invasive (EEG) and invasive (sEEG) modes and is applied to multiple mature benchmark datasets. Figure 7 provides excellent qualitative evidence by comparing the learned functional regions with known brain emotion processing centers, enhancing the biological rationale for the segmentation mechanism and demonstrating the interpretability of the framework. Furthermore, the framework exhibits strong robustness to changes in topology; Figure 6 effectively demonstrates its resistance to random channel loss, showing a more gradual performance degradation compared to robust baseline models such as mdJPT and BaRISTA.

**Weaknesses:**
The computational efficiency of the block sparse attention module assumes a balanced channel distribution, but real-world sEEG data is highly imbalanced. This paper does not explore how this imbalance affects hardware efficiency, nor does it explain whether padding introduces additional overhead. Although the top-p gating mechanism was removed, the achieved sparsity was not analyzed to examine its impact on memory and runtime. Furthermore, while the pass-through Gumbel-Softmax estimator may lead to training instability, no loss curves or variance statistics are provided to validate the stability of the optimization. Finally, the lack of sensitivity analysis for key hyperparameters (e.g., temperature $\tau$ and annealing policy $\alpha(t)$) obscures the difficulty of tuning these parameters, hindering the reproducibility of the results and their practical application.

---

> ### Author Rebuttal · Authors · 2026-03-31
>
> We thank the reviewer for the thoughtful feedback and refer you to the rebuttal supplement (Tables S1–S17, Figs. S1–S7): https://anonymous.4open.science/r/ICML2026_Rebuttal-6FDE/icml2026_rebuttal.pdf
> ## Re: W1, Q3
> We agree that the balanced-$P$ complexity analysis in Appendix B.1 is an idealized simplification, while real sEEG is imbalanced across regions and subjects. We therefore added Table S17, which measures the effect of this imbalance on attention efficiency for real sEEG region-channel distributions. The imbalance ratio ranges 2.17-5.04, confirming the balanced assumption is not met in practice. Nevertheless, RECTOR-SA remains faster than dense channel-time attention for every subject, with empirical speedups ranging from 1.07×-3.10× and a 2.81× speedup under mixed-subject batching in practice
>
> We would like to clarify that we do not use padding for attention in our implementation. In the mixed-subject setting, each batch contains samples from 4 subjects, but attention is computed separately per subject, and local functional attention is executed exactly on each non-empty region block. Empirically, this block execution is consistently better than padded execution in both runtime and memory. Overall, region imbalance reduces the idealized balanced-case speedup, but does not negate the RECTOR-SA's practical hardware efficiency on sEEG
> ## Re: W2, Q2
> We would like to clarify that top-$p$ gating is primarily intended as an adaptive denoising mechanism rather than the main source of computational savings. In RECTOR, the dominant efficiency gain comes from the block-sparse RECTOR-SA. By contrast, as written in Eqs. 3, 6, and 7, top-$p$ is applied after the logits are formed, so its isolated effect on runtime/memory can be small, and its main value is adaptive filtering within the already block-sparse attention structure
>
> We use Table S14 mainly to quantify its empirical sparsity: top-$p$ removes a substantial fraction of low-salience candidate connections (44.9–57.2% for local functional attention and 36.3–41.7% for global), supporting that it is actively filtering noisy interactions. This matches the -top-$p$ ablation in Table 3, where removing top-$p$ degrades downstream performance, indicating that its main benefit is representation quality via adaptive denoising
> ## Re: W4, Q1
> We have now added explicit sensitivity analyses for the annealing policy $\alpha(t)$ and the Gumbel-Softmax temperature $\tau$. Across EEG/sEEG benchmarks, RECTOR is not sensitive to nearby choices, but it does benefit from gradual annealing. In Tables S6–S7, the default cosine decay consistently gives the best/near-best performance, while constant $\alpha(t)=0$ is clearly worse and constant $\alpha(t)=1$ or a hard $1\rightarrow 0$ step remain competitive but usually underperform gradual decay. This shows that immediately removing the anatomical prior or keeping it fixed is suboptimal, whereas progressively relaxing it yields better final accuracy. Qualitatively, Fig. S5 shows the same trend: constant $\alpha(t)=1$ keeps the learned partition overly anatomy-constrained, the hard $1\rightarrow 0$ step allows adaptation but gives a less smooth partition, and cosine annealing produces the most coherent functional refinement. Overall, gradual schedules outperform degenerate ones, but performance differences among all reasonable schedules are modest, indicating low tuning difficulty in practice
>
> We also added temperature ablations in Tables S8–S9: the default two-stage schedule $0.5\rightarrow 1.0$ achieves the best/near-best results across datasets, indicating that RECTOR is robust to nearby temperature settings while slightly benefiting from sharper early assignments and smoother later optimization
>
> More broadly, Tables S1–S2 and Fig. S1 show that performance remains competitive under moderate changes in the number of regions, suggesting that AFP does not require delicate hyperparameter tuning to work well in practice
> ## Re: W3
> We have added both optimization trajectories and seed-wise variance statistics to directly assess training stability. In Fig. S7, the total pretraining loss decreases smoothly for EEG (SEED) and sEEG (MSIT) and all unweighted loss components remain well behaved, with narrow shaded std bands across 5 random seeds and no evidence of significant instability, collapse, or divergence. Importantly, the topology-related term $\mathcal{L}_\text{TSM}$, which is most directly affected by the Gumbel-Softmax in AFP, also decreases steadily, suggesting that the estimator does not introduce practical optimization instability
>
> Tables S10–S11 report downstream variance across random seeds, showing that this stable pretraining behavior carries through to downstream task: the stds are very small on SEED/SEED-IV (0.39-0.56) and MSIT/ECR (0.33-0.42), and modest on DEAP (1.54-1.93). Together, these results indicate that RECTOR’s optimization is stable across random initializations despite the use of a Gumbel-Softmax estimator

---

> > ### Author Rebuttal · Reviewer_Ze3F · 2026-04-02
> >
> > I appreciate the authors' efforts to address my comments. My question has been largely resolved, I will consider adjusting my score.

---

> > > ### Author Response · Authors · 2026-04-03
> > >
> > > We sincerely appreciate your dedicated time and constructive evaluation. We are very glad that our response successfully addressed your questions, and we are grateful for your positive recommendation and for raising your score.
> > >
> > > Here is a summary of our new analyses:
> > > 1. **Efficiency under channel imbalance**: We added measurements on real imbalanced sEEG region-channel layouts, clarified that attention is executed exactly per subject/region without padding, and showed that RECTOR-SA still provides consistent speedups over dense attention in practice.
> > > 2. **Empirical sparsity of top-$p$ gating**: We quantified the fraction of pruned local/global connections and clarified that top-$p$ mainly serves as an adaptive denoising mechanism, improving representation quality more than runtime.
> > > 3. **Hyperparameter robustness and training stability**: We added both quantitative and qualitative sensitivity analyses for the annealing policy, temperature, and number of regions, along with multi-seed loss curves and variance statistics, showing stable training and low sensitivity to reasonable settings.
> > >
> > > We are grateful for your help in refining the technical rigor of this manuscript, and we hope these newly incorporated analyses are helpful for your final evaluation.

---

### Official Review · Reviewer_yJ5f · 2026-03-15

**Soundness:** 2
**Presentation:** 3
**Significance:** 3
**Originality:** 3
**Overall Recommendation:** 4
**Confidence:** 3

**Summary:**

This paper proposes RECTOR, a self-supervised framework for EEG/sEEG representation learning that tries to jointly model regions, channels, and time, instead of treating EEG as just a flat sequence of channel-time tokens. The method is evaluated on several EEG emotion recognition datasets and two sEEG task engagement datasets, with additional results on missing-channel robustness, cross-montage transfer, and some interpretability analysis.

**Compliance With Llm Reviewing Policy:**

Affirmed.

**Final Justification:**

I maintain my score

**Key Questions For Authors:**

What exactly should I think of AFP-learned regions as? Are these meant to be interpreted as a form of task-dependent functional parcellation over electrodes, or more as an internal grouping mechanism for attention?

How stable are the learned region assignments?
For example, across random seeds, across subjects, or across sessions, do the AFP partitions converge to similar structures, or are they highly variable?

How much of the gain comes from better structure, and how much comes from simply having a more favorable inductive bias than the baselines?
The ablations are helpful, but I would still like a bit more discussion here, especially since Full SA removes AFP and LTSM together. If there is a cleaner comparison isolating sparse structured attention from the topology objective, that would be better.

**Limitations:**

I would suggest adding a short but concrete discussion of the limits of interpreting learned functional regions as neuroscience findings.

**Strengths And Weaknesses:**

Strengths

The architecture and the SSL objective are actually designed to match each other. AFP defines adaptive channel-to-region assignments, RECTOR-SA uses them to build sparse hierarchical attention, and TSM explicitly regularizes those assignments in latent space.

They do not stop at reconstruction, but add cross-view consistency and topology modeling. That makes the framework feel more substantial.

The method is evaluated on multiple EEG and sEEG benchmarks, and the gains are not tiny. The ablations are reasonably comprehensive.


Weaknesses

My main concern is that the paper’s conceptual language is sometimes stronger than the evidence. The paper talks a lot about “functional topology,” “adaptive functional regions,” and “neurophysiologically plausible representations,” but most of the supporting evidence comes from attention maps, class activation maps, RIS curves, and ablations. These are useful, but they are still indirect. I do not think the paper fully shows that the learned regions correspond to stable or meaningful functional units in a strong neuroscientific sense.


I'd like to see a more direct comparison between learned AFP partitions and fixed anatomy beyond downstream performance and robustness. The paper shows that AFP helps, but it is still a bit unclear what exactly is being learned, how stable those learned partitions are across runs or subjects, and how much they really deviate from the initialization in a meaningful way.


One more small concern is about interpretability analysis. The paper shows channel-level class activation maps and gives a plausible frontal asymmetry story, but the actual procedure for generating those explanations is not described very clearly in the main paper. So I found the neuroscience interpretation interesting, but not fully convincing yet.

---

> ### Author Rebuttal · Authors · 2026-03-31
>
> We thank the reviewer for the thoughtful feedback and refer you to the rebuttal supplement (Tables S1–S17, Figs. S1–S7): https://anonymous.4open.science/r/ICML2026_Rebuttal-6FDE/icml2026_rebuttal.pdf
> ## Re: W1, W2, Q2
> We agree our claim should be supported by more direct evidence on what AFP learns and how stable it is. We therefore added qualitative comparisons of the learned partitions. Fig. S1 shows AFP does not simply preserve the anatomical initialization. From anatomical priors, the learned partitions merge and split initial regions into smoother data-driven groupings: nearby channels separated can be reassigned into the same region, while anatomically broad regions are divided into refined substructures. The learned partitions remain spatially contiguous. This indicates structured functional refinement of the prior. The 9-region random initialization further supports this: when the initialization lacks anatomical structure, the learned organization is less coherent and performs substantially worse (Tables S1–S2), suggesting that AFP meaningfully reshapes the prior
>
> Fig. S4 shows the learned partitions across random seeds, sessions, and subjects. AFP is highly stable across seeds and sessions, with the same macro-scale organization, while cross-subject variation is larger but still structured. This matches Table S5, where variability is lowest across seeds (\~3%), slightly higher across sessions (\~5%), and highest across subjects (>10%). The learned assignments converge to similar large-scale topologies, with subject-specific differences mainly as localized refinements. This supports that AFP learns reproducible, spatially meaningful partitions beyond downstream performance. For more evidence that AFP learns structured refinements beyond fixed anatomical or neuroscience priors, see our response to Reviewer YWsS W2, Fig. S6, and Table S16
> ## Re: Q1, Limitations
> We view AFP-learned regions primarily as data-driven, task-adaptive grouping variables for structured attention, rather than as a claim of definitive neuroscientific parcellation. They are not arbitrary internal clusters: the learned assignments are anatomy-initialized, remain spatially coherent, and are consistent across seeds and sessions, with more modest structured variation across subjects
>
> Thus, the accurate interpretation is: AFP learns a task-dependent, neurophysiologically plausible functional partition over electrodes that serves as the grouping structure for RECTOR-SA. These “functional regions” are best understood as more meaningful than technical attention grouping, but weaker than cortical parcellation in a strong neuroscientific sense. This is also consistent with Fig. S1, showing structured deviations from anatomical initialization rather than arbitrary reassignment. We will add this discussion to the Limitations section.
> ## Re: W3
> We agree that the procedure for generating channel-level explanations in Fig. 7(b) should be described more clearly in the main paper rather than in Appendix F.3. Concretely, these maps are generated using Grad-CAM [1, see our responses to Reviewer qUWE, Reference]: after forwarding an input trial through finetuned RECTOR, we apply Grad-CAM before the classification head to obtain a per-channel importance score for the class. The visualization is obtained by averaging these scores across subjects. Thus, Fig. 7(b) is based on a Grad-CAM attribution procedure rather than visual inspection. We will move this description to the main paper in revision
> ## Re: Q3
> We agree that the cleanest interpretation of Table 3 is to separate the contribution of better structure from the inductive bias. Here, “better structure” corresponds to AFP+TSM, while the “favorable inductive bias” corresponds to RECTOR-SA. These components are not independent by construction: if RECTOR-SA is replaced with Full SA (dense channel-time attention), region assignments are no longer needed, so AFP and TSM are removed together; similarly, once AFP is removed, the model reverts to fixed anatomical grouping, so there is no assignment variable for TSM to supervise. Thus, a perfectly orthogonal ablation is not available because the topology objective is only defined when adaptive assignments exist
>
> Under this interpretation, the -AFP ablation in Table 3 isolates the value of learned structure (AFP+TSM) relative to fixed anatomy, whereas the gap between -AFP and Full SA reflects the additional contribution of the structured inductive bias. Empirically, -AFP is consistently better than Full SA across datasets (e.g., SEED w-F1 71.6 vs 70.2), showing even with fixed anatomy, RECTOR-SA already improves over dense attention. The full model further improves over -AFP (e.g., SEED w-F1 73.1 vs 71.6), indicating that AFP-learned topology and topology modeling contribute additional gains beyond the inductive bias. Thus, the gains come from both sources: RECTOR-SA provides the structured inductive bias, and AFP+TSM further improve the learned structure

---

> > ### Author Rebuttal · Reviewer_yJ5f · 2026-04-04
> >
> > Thanks for the rebuttal. I'll maintain my positive initial score.

---

> > > ### Author Response · Authors · 2026-04-06
> > >
> > > Thank you so much for your dedicated time and constructive feedback. We are very glad that our response fully resolved your concerns, and we sincerely appreciate your continued positive recommendation.
> > >
> > > To briefly summarize the main additional analyses:
> > > 1. **AFP partitions: interpretation, structure, and stability**: We clarified that AFP-learned regions are best viewed as task-adaptive grouping variables for structured attention rather than definitive neuroscientific parcellation, and we added new qualitative and quantitative analyses showing that AFP learns structured refinements beyond anatomy that remain spatially coherent and stable across random seeds, sessions, and subjects; and the weaker random-initialization control further indicates that AFP meaningfully reshapes the prior.
> > > 2. **Inductive bias vs. better structure**: We expanded the discussion of Table 3 to clarify that the gains come from both sources: the structured inductive bias of RECTOR-SA and the additional learned topology introduced by AFP and TSM beyond fixed anatomy.
> > > 3. **Interpretability procedure**: We clarified the Grad-CAM procedure used to generate the channel-level explanations in Fig. 7(b).
> > >
> > > We deeply appreciate your help in strengthening the technical rigor of this manuscript, and we hope these clarifications and additional analyses are helpful for your final evaluation.

---

### Official Review · Reviewer_YWsS · 2026-03-24

**Soundness:** 4
**Presentation:** 3
**Significance:** 3
**Originality:** 3
**Overall Recommendation:** 4
**Confidence:** 3

**Summary:**

RECTOR is a self-supervised framework for EEG/sEEG representation learning. It learns functional channel-to-region assignments (AFP) that drive a block-sparse hierarchical attention (RECTOR-SA), and trains with a multi-objective SSL scheme (MTRL) combining masked prediction, topology modeling, and cross-view consistency. Evaluated on three EEG emotion benchmarks and two sEEG cognitive-state datasets, with strong results across all of them.

**Compliance With Llm Reviewing Policy:**

Affirmed.

**Final Justification:**

I maintain my score. The follow-up analyses address my remaining questions, and my concerns are resolved.

**Key Questions For Authors:**

Q1. How sensitive is performance to the number of regions R? Do the learned partitions change qualitatively at different granularities?

Q2. It is not clear how AFP is handled during fine-tuning. Are the assignment parameters (W, P) frozen or updated? If updated, does the learned topology shift significantly from what was learned during pre-training?

Q3. The current evaluation covers emotion and cognitive-state classification. Would RECTOR generalize to other EEG paradigms (e.g., motor imagery, seizure detection) or other task types beyond classification?

Q4. Have you tried pre-training on one dataset and evaluating on a completely unseen one? If that works, it would significantly strengthen the case for RECTOR as a general-purpose EEG representation model.

**Limitations:**

The paper does not explicitly discuss its own limitations. It would be worth acknowledging modeling assumptions such as the hard one-hot channel-to-region assignment, which does not capture cases where a channel participates in multiple functional networks.

**Strengths And Weaknesses:**

## Strengths

The architecture and training objectives are well tailored to the characteristics of EEG data, and the SSL components are appropriately chosen to support the proposed structure. The framework has many components, but the ablation study is thorough enough to justify each one, and the experiments include clinically relevant scenarios like sensor dropout and cross-montage transfer. Computational cost is also properly addressed, with a detailed complexity analysis backed by empirical training time comparisons. The appendix provides sufficient implementation and training details for reproduction.

## Weaknesses

The ablation covers each module well, but I would have liked to see more analysis around AFP's design choices. The number of regions R, the annealing schedule $\alpha(t)$, and the loss weights are all fixed based on convention or tuning, but their sensitivity is not explored. Since AFP is what distinguishes RECTOR from prior work, understanding how these choices shape the learned partitions would be valuable. The interpretability analysis confirms that AFP recovers known structure (e.g., frontal asymmetry), which is reassuring, but it would be more compelling if the learned partitions also revealed groupings not already expected from neuroscience literature. Also, the current AFP produces a static partition per dataset, while brain functional connectivity is known to shift over time. Whether the partition could vary within a trial seems worth exploring.

There are also several minor typos:
- "schemes learns" $\rightarrow$ "schemes learn" (Section 1)
- "d-dimentional" $\rightarrow$ "d-dimensional" (Section 2.1)
- "Hyperparemeter" $\rightarrow$ "Hyperparameter" (Table 10, 11)
- "RECTOR-Tramsformer" $\rightarrow$ "RECTOR-Transformer" (Appendix B.5)
- Figure 7 caption: panel numbering goes (a), (b), (c), (4), should be (d)
- Algorithm 2: (e) might be (d)

---

> ### Author Rebuttal · Authors · 2026-03-30
>
> We thank the reviewer for the thoughtful feedback. We will correct the mentioned typos and we refer you to rebuttal supplement (Tables S1–S17, Figs. S1–S7): https://anonymous.4open.science/r/ICML2026_Rebuttal-6FDE/icml2026_rebuttal.pdf
> ## Re: W1, Q1
> We added sensitivity analyses for AFP’s design choices, focusing on the number of regions $R$ and $\alpha(t)$; the loss weights were selected by validation-based hyperparameter tuning
>
> For $R$, Tables S1–S2 show that RECTOR is not highly sensitive to changes in granularity: performance remains competitive across a broad range of $R$, while the default is generally best/near-best. For EEG, the gap between $R=6,9,12$ is modest, whereas very coarse/fine partitions are slightly worse, suggesting AFP benefits from intermediate granularity rather than delicate tuning. Importantly, the 9-region random initialization is clearly worse than anatomical-prior, showing that performance depends on meaningful partition structure rather than region count alone
>
> Fig. S1 shows that smaller $R$ yields coarser macro-regions and larger $R$ finer subdivisions, while spatial organization remains stable and neurophysiologically plausible. Thus AFP does not produce arbitrary partitions when $R$ changes; it refines the same large-scale structure at different levels of detail
>
> We also added sensitivity analyses for $\alpha(t)$ in Tables S6–S7 and Fig. S5. RECTOR is again not highly sensitive, but gradual cosine annealing yields the most coherent topology and best/near-best performance; For further discussion, see our response to Reviewer Ze3F, W4 Q1
>
> Overall, AFP is robust to reasonable hyperparameter choices while still benefiting from gradual topology refinement
> ## Re: Q2
> In finetuning, AFP is updated and the assignment parameters are not frozen. Table S3 shows that only 5–7% of channels change region from pretraining to finetuning. Fig. S2 shows the same pattern: the macro-scale partition is largely preserved, and changes are local rather than wholesale. Thus, finetuning mainly performs task-specific local refinement on top of the pretrained topology
> ## Re: W2
> We added Fig. S6 and Table S16 comparing the anatomical prior with the learned AFP partitions. A substantial fraction of channels is reassigned from anatomy (36.5–51.5%), showing that AFP is not trivially recovering the anatomical grouping; on SEED/SEED-IV, these reassignments are also highly consistent across sessions (84.7%/86.1%)
>
> Fig. S6 shows that these deviations are spatially coherent and reproducible. Beyond frontal-asymmetry-related changes, AFP consistently refines anatomy by forming a stable anterior subgroup (Fz/F2/F4), separating nearby fronto-central electrodes (F1/FC3/FC1/FCz/Cz), regrouping a left lateral chain (FC5/C5/CP5) with left temporal electrodes, and making posterior parietal-occipital organization more asymmetric. These patterns suggest AFP is not merely recovering known anatomy, but refining it into stable, task-relevant functional groupings that may help neuroscientists generate more hypotheses about channel organization
> ## Re: W3
> We would clarify that AFP does not produce a static partition per dataset. It is data-adaptive: assignments are inferred from each input sample (Eq. 4), so functional grouping can vary over time
>
> To examine it, we added a within-trial analysis in Fig. S3/Table S4 comparing neighboring samples in same trial. AFP allows temporal variation, but it is structured and limited rather than arbitrary: only a small fraction of channels changes assignment between adjacent samples (pretraining: 3.3–6.1%; finetuning: 2.1–4.4%). Fig. S3 shows that the large-scale partition is preserved across neighboring samples, while only a small subset of channels undergoes local reassignment. Thus, AFP adapts within a trial in a stable way and preserves macro-scale organization while allowing local refinement
> ## Re: Q3
> We added Table S15 on PhysioNet-MI motor imagery classification. RECTOR achieves the best performance (65.5 w-F1/65.6 BAcc), outperforming both task-specific MI models (ATCNet, TMSA-Net) and general baselines. It indicates that RECTOR is generally effective across distinct EEG paradigms and neurophysiological structures. The core design of RECTOR can in principle be paired with other task heads, and we view extension to non-classification settings as future work
> ## Re: Q4
> We added cross-dataset transfer results in Tables S12–S13 for pretraining on one dataset and finetuning on an unseen one. It consistently improves over training from scratch (e.g. SEED, pretrain on unseen SEED-IV improves 77.2 to 84.2 in w-F1, SD). As expected, pretraining on the target dataset remains best, but the gains over scratch by cross-dataset pretraining support RECTOR as a more general EEG/sEEG representation model. Broader transfer across heterogeneous paradigms is also reflected in Fig. 6(b)
> ## Re: Limitation
> We will add this limitation and frame overlapping region memberships as an important future direction

---

> > ### Author Rebuttal · Reviewer_YWsS · 2026-04-04
> >
> > Thank you for the additional experiments. Most of my concerns are addressed, though a couple of points remain.
> >
> > The non-anatomical groupings are shown to be stable and spatially coherent, but stability alone does not establish functional significance. Is there a way to validate these groupings against an independent measure beyond task accuracy? Along the same lines, knowing whether the 5-7% of channels that shift during fine-tuning correspond to task-relevant changes or noise would strengthen the task adaptation argument for AFP.

---

> > > ### Author Response · Authors · 2026-04-06
> > >
> > > We sincerely thank the reviewer for this insightful follow-up. We are glad our clarification has addressed most concerns and agree that stability alone may not be sufficient to show functional significance; accordingly, we added two new analyses beyond downstream accuracy and refer you to the updated rebuttal supplement (Tables S18–S19): https://anonymous.4open.science/r/ICML2026_Rebuttal-6FDE/icml2026_rebuttal.pdf
> > >
> > > ### **1. Functional significance of the learned non-anatomical groupings beyond task accuracy**
> > > Table S18 provides a representation-level independent measure beyond downstream task accuracy to validate the functional significance of the learned non-anatomical groupings. We evaluate the pretrained region representations before finetuning, using **intra-class distance** (Intra-cls dist.), **inter-class distance** (Inter-cls dist.), and **Silhouette score**. For each sample we extract the frozen region representations flattened to $z\in\mathbb{R}^{Rd}$, where $R$ is the number of regions. Distances are computed with cosine distance between samples. Intra-cls dist. is the average distance over pairs with the same class, while Inter-cls dist. is the average over pairs with different classes. The Silhouette score evaluates how well the frozen representations form label-consistent structure by jointly measuring within-class compactness and between-class separation; higher values indicate better class separation
> > >
> > > Across all EEG/sEEG datasets, AFP with anatomical initialization consistently yields lower Intra-cls dist., higher Inter-cls dist. and Silhouette score than both fixed anatomy and AFP with random initialization (e.g., SEED: Intra-cls 0.563→0.521, Inter-cls 0.903→0.939, Silhouette 0.400→0.423 vs fixed anatomy). It shows that the AFP-learned non-anatomical groupings have task-relevant functional significance: the AFP-learned topology organizes the pretrained representation into more label-consistent and better-separated latent structure, even before finetuning and thus beyond downstream accuracy
> > > ### **2. The channel shifts during finetuning are task-relevant adaptation**
> > > To test whether the 5–7% of channels shifted during finetuning reflect task-relevant adaptation rather than noise, Table S19 adds two complementary analyses
> > >
> > > (1) We mask the reassigned (“Shifted”) channels at inference and measure the **relative true-class logit-margin drop** $\Delta m$. Let $\ell_k(x)$ be the class-$k$ logit of the finetuned classifier for input $x$, and $y$ the ground-truth label of $x$; we define the true-class logit margin of $x$:
> > > $$m_y(x)=\ell_y(x)-\max_{k\neq y}\ell_k(x)$$
> > > We then define the relative true-class logit-margin drop:
> > > $$\Delta m(x)=\frac{m_y(x)-m_y(x_\text{masked})}{|m_y(x)|+\epsilon}$$ where $x_\text{masked}$ is the input after masking the shifted channels, and $\epsilon$ ensures numerical stability. Higher $\Delta m$ means that masking shifted channels causes a larger relative reduction in the model’s ability to separate the correct class from its strongest competitor, so the masked channels are more task-relevant. In every dataset, masking the shifted channels in the AFP-active model causes a substantially larger drop than masking either a random matched subset or the same shifted indices in an AFP-frozen control (e.g., DEAP-A-SD: $\Delta m$ = 12.4\% vs 7.3% random, 6.6% frozen)
> > >
> > > (2) We quantify the **Gini coefficient of the normalized region importance score (RIS) distribution** $G_\text{RIS}$. If $\{p_1,\dots,p_R\}$ are the RISs with $\sum_r p_r=1$, then $G_\text{RIS}$ is the Gini coefficient of $\{p_r\}$, where a higher value indicates more concentrated regional attribution. AFP-active consistently yields higher $G_\text{RIS}$ than AFP-frozen (e.g., DEAP-A-SD: 57.9% vs 52.4%), indicating that finetuning with adaptive AFP concentrates task-relevant attribution into a smaller set of dominant regions. Together, these results support that the small reassignment set during finetuning reflects selective task-relevant topological refinement
> > >
> > > **Table S18 shows that the AFP-learned non-anatomical groupings are functionally meaningful beyond downstream accuracy**, while **Table S19 shows that the limited finetuning shifts are selectively task-relevant**. Together with the previously demonstrated stable and spatially coherent partitions, these results confirm that AFP learns reproducible, task-relevant non-anatomical groupings and that finetuning induces selective topological adaptation. We do not claim that these analyses constitute definitive external neuroscientific validation; rather, they provide accuracy-independent evidence that the learned groupings and the finetuning shifts are structured and task-relevant within the learned representation. We will revise this accordingly in the manuscript
> > >
> > > We are grateful for your help in refining the technical rigor of this manuscript, and hope these clarifications and new results address your follow-up concerns and are helpful for your final evaluation

---

### Decision · Program_Chairs · 2026-04-30

**Decision:**

Accept (regular)

**Comment:**

This paper proposes a self-supervised framework for EEG and sEEG representation learning that combines adaptive functional partitioning, block-sparse hierarchical attention, and a multi-objective SSL training scheme to jointly model region, channel, and temporal structure. Reviewers generally found the paper technically strong, well motivated, and thoroughly evaluated, highlighting its consistent gains across multiple EEG and sEEG benchmarks, careful ablations, robustness analyses for missing channels and cross-montage transfer, and solid treatment of computational efficiency and reproducibility. While some questions remained about the neuroscientific strength of the interpretability claims, sensitivity to several hyperparameters, and the role of the initial anatomical prior, the consensus was positive, and the rebuttal appears to have resolved most of the substantive concerns.